# A tale of two goals: leveraging short term goals performs best in multi-goal scenarios

**Olivier Serris**                                                    *Olivier.Serris@isir.upmc.fr*
*ISIR*
*Sorbonne Université, CNRS*

**Stéphane Doncieux**                                          *Stephane.Doncieux@isir.upmc.fr*
*ISIR*
*Sorbonne Université, CNRS*

**Olivier Sigaud**                                                  *Olivier.Sigaud@isir.upmc.fr*
*ISIR*
*Sorbonne Université, CNRS*

**Reviewed on OpenReview:** *https://openreview.net/forum?id=qsUeLwbErp*

## Abstract

When an agent must learn to reach far away goals, several hierarchical reinforcement learning methods leverage planning to create a sequence of intermediate goals that guides a lower-level goal-conditioned policy. The low-level policy is typically conditioned on the current goal, with the aim of reaching it as quickly as possible. However, this approach can fail when intermediate goals can be reached in multiple ways, some of which may prevent reaching subsequent goals. To address this issue, we introduce an enriched Markov Decision Process (MDP) framework in which the optimization objective not only considers reaching the current goal, but also subsequent ones. Using this framework, we can specify the goals that the agent prepares to achieve in advance. To study the impact of this design, we perform a series of experiments on navigation, balance and locomotion tasks in which sequences of intermediate goals are given. By evaluating policies trained with an off-policy actor-critic algorithm on both the standard goal-conditioned MDP framework and ours in these tasks, we show that learning policies conditioned on the next two goals generally require less interaction data than all other types to reach the same level of performance or better.

## 1 Introduction

In reinforcement learning (RL), an agent learns to control a system to accomplish a task over a sequence of actions by maximizing some reward signal. Standard RL methods struggle when the sequence of actions is long and the reward signal is sparse. Goal-conditioned (GC) agents (Schaul et al., 2015), where the agent must learn a GC-policy to reach any goal in a given goal space, are not exceptions, often failing to reach distant goals. To address this, several hierarchical reinforcement learning (HRL) approaches propose to decompose the task into a sequence of intermediate goals, with the GC-policy iteratively conditioned on each goal until the final one is reached (Eysenbach et al., 2019; Levy et al., 2019).

However, as shown in Chenu et al. (2023), when intermediate goals do not fully specify the states in which they are achieved, a *chaining* issue can arise: the system may achieve a goal in a state that is incompatible with the achievement of the next goal. To counteract this, Chenu et al. (2023) propose a framework in which the agent prepares to reach future goals while aiming at the current one. Their solution consists in integrating the successive goals into the state space of the underlying Markov Decision Process (MDP) together with adapting the transition and reward functions. This results in a *sequential goal reaching* MDP,

but their approach is limited to the case where the agent must learn to follow a unique predefined sequence of goals from a given starting state.

In this paper, we address the much more general context in which the agent must learn to achieve any final goal from any starting state given a sequence of intermediate goals. In this context, the GC-policy must be able to reach the current goal in a way compatible with the rest of the sequence which itself depends on the final goal. Moreover, while learning to reach a goal, it must maintain its ability to reach any other goal from any other sequence.

Drawing inspiration from Chenu et al. (2023), we propose a Markov Decision Process (MDP) in which the state, transition and reward functions account for sequential goal reaching and make it possible to fully benefit from goal relabeling provided by an adapted version of the Hindsight Experience Replay (HER) mechanism (Andrychowicz et al., 2018). Building on this framework, we focus on learning the low-level GC-policy and assume access to an expert high-level planner that supplies the sequence of intermediate goals given a start state and a final goal. We propose an RL algorithm, GCP-$N$, in which the agent is conditioned on both the first and the $N$-th goal. In this formulation, the agent optimizes its actions to reach all intermediate goals up to the $N$-th one.

Then, to evaluate these instances, we implement three agents on top of the same actor-critic architecture: an agent conditioned only on the final goal, a myopic agent that only learns to reach the current goal before discovering the next one, and our GCP-$N$ solution. We compare these agents on various navigation, balance and locomotion benchmarks in which the agent is trained to reach any goal from any state given some well-chosen sequence of intermediate goals.

In summary, our contributions are the following: (1) we investigate the failure modes of myopic agents, relating them to the terminal conditions of the underlying MDP; (2) we propose a new MDP formalization which accounts for sequential goal reaching; (3) we propose the GCP-$N$ method, which combines Soft Actor-Critic (SAC) with a modified version of HER adapted for multi-goal conditioning; (4) Given the possibility of preparing up to $N$ goals in advance, we compare how different choices of $N$ affect different performance criteria: the episode return, the success rate or the time to goal, depending on the environment.

Our results suggest that policies conditioned on the two most immediate successive goals ($N = 2$) can generally be learned from less interaction data than myopic agents and non-sequential agents to reach the same level of performance or better. Our analysis shows that the GCP-2 agent benefits from the simplicity of learning short-horizon objectives while still being able to complete the full sequence of goals.

## 2 Background

In this work, we study goal-conditioned reinforcement learning agents that leverage hindsight relabeling and can be seen as the low-level part of a hierarchical approach. In this section, we provide background information about these elements and present some related algorithms.

### 2.1 Reinforcement learning

In reinforcement learning (RL), the interactions of an agent with an environment are defined as a Markov Decision Process (MDP) $\mathcal{M} = (S, A, R, P, \gamma, \rho_0, \mathcal{T})$, where $S$ is the state space, $A$ is the action space, $R$ is the reward function, $P$ is the transition probability function, $\gamma$ is the discount factor and $\rho_0$ is the initial state distribution. In the episodic case, at the beginning of each episode, an initial state $s_0$ is drawn from the initial state distribution $\rho_0$. Then at each step, the agent observes $s_t \in S$, chooses an action $a_t \in A$, moves to a new state $s_{t+1}$ according to $P$ and receives a reward $r_t$ from $R$. Finally, $\mathcal{T}$ corresponds to the set of all terminal states. The goal of RL is to learn an agent policy $\pi_\theta$ that maximizes the expected cumulative reward until a terminal state is reached, that is

$$\mathbb{E}_{\substack{a \sim \pi_\theta \\ s_0 \sim \rho_0}} \left[ \sum_{t=0}^{\infty} \gamma^t r_t \prod_{i=0}^{t} 1 - \mathbb{1}[s_i \in \mathcal{T}] \right].$$

## 2.2 Goal-conditioned reinforcement learning

Goal conditioned reinforcement learning (GCRL) can be formalized as an extension of MDPs that we call a goal-conditioned MDP (GC MDP) and that we note $\mathcal{M}_{gc} = (S_{gc}, A, R_{gc}, P, \gamma, \rho_0, \mathcal{T})$, with $S_{gc} = S \times G$. Now, at the beginning of each episode, a single goal $g \in G$ is selected in addition to $s_0$. A goal specifies a state or a set of states. For example, a goal might correspond to the 3D position of the center of mass of a robot, and achieving the goal could mean reaching any position within a sphere of diameter $\epsilon$ around that point. A mapping function $\phi : S \to G$ can project a state into the goal space. At each step, the agent now observes $(s_t, g) \in S_{gc} = S \times G$ and receives a reward $r_t$ from the goal-conditioned reward function $R_{gc}$. In particular, if $S_g$ is the set of all states for which $g$ is achieved, a sparse reward function is defined as: $R_{gc}(s_t, g_t, s_{t+1}) = \mathbb{1}[s_{t+1} \in S_{g_t}]$.

Finally, $\mathcal{T}(g)$ defines the set of terminal states based on the current goal $g$. Some states may be terminal independently from the current goal (e.g. a robot falling on the floor). Reaching a goal can be terminal or not depending on the task at hand. For instance, in a maze environment, the task ends as soon as the agent reaches the goal position, while in a pole-balancing task, the pole must stay at equilibrium in the goal state until the episode ends, hence goals are not terminal.

The objective of GCRL algorithms is to find a GC-policy $\pi(s, g)$ that maximizes the expected cumulative reward:

$$\mathbb{E}_{\substack{a \sim \pi_\theta \\ s_0 \sim \rho_0 \\ g \sim P(g)}} \left[ \sum_{t=0}^{\infty} \gamma^t R_{gc}(s_t, g, s_{t+1}) \prod_{i=0}^{t} 1 - \mathbb{1}[s_i \in \mathcal{T}(g)] \right]. \tag{1}$$

A common way to tackle the GCRL objective is through actor-critic methods. These algorithms learn a critic, which estimates a goal-conditioned action-value function $Q(s, a, g)$, measuring the expected return of taking action $a$ in state $s$ while pursuing goal $g$, and then following the current policy. The actor, represented by the policy $\pi(s, g)$, is updated to maximize the expected value given by the critic. Off-policy algorithms are particularly useful in GCRL, as they can reuse past experience across different goals. In practice, both the actor and the critic are trained using transitions $(s_t, g, a_t, r_t, s_{t+1},)$ sampled from a replay buffer, which stores past experience collected by the agent. In this article, we use Soft Actor-Critic (SAC) (Haarnoja et al., 2018), which augments the actor-critic framework with an entropy term to encourage exploration.

## 2.3 Hindsight relabeling

GCRL with sparse reward functions is difficult because, as long as the agent does not reach the goal, it does not collect any reward and does not learn anything. The Hindsight Experience Replay (HER) algorithm (Andrychowicz et al., 2018) is designed to mitigate this difficulty. During off-policy updates, where the agent learns from transitions $\{s_t, g, a_t, R_{gc}(s_t, g, s_{t+1}), s_{t+1}\}$, most of the time it does not receive a reward. HER modifies transitions by substituting the current goal with a goal achieved later in the trajectory, resulting in a more dense reward signal.

This is shown in the following notation, where $t_{max}$ represents the last step of a given trajectory:

$$(s_t, \cancel{g, R_{gc}(s_t, g, s_{t+1})}, a_t, s_{t+1}) \to (s_t, \phi(s_k), R_{gc}(s_t, \phi(s_k), s_{t+1}), a_t, s_{t+1}), \text{where } k \in [t, t_{max}]. \tag{2}$$

## 2.4 Sequential planning in GCRL and myopic policies

When the agent's goal is too far from its current state, combining GCRL with HER is not enough. In that case, established approaches in the HRL literature often combine a GC-policy together with a planner. We formalize three main points. (1) How a planner outputs plans, (2) How plans are updated and (3) How to combine a planner with a GC-policy.

We define the planner as a function that outputs a sequence of intermediate goals. For $i \geq 1$, let $G^i := \underbrace{G \times G \times \cdots \times G}_{i \text{ times}}$ denote the set of all sequences of length $i$ over $G$. We then define $G^+ := \bigcup_{i \geq 1} G^i$, so that

$G^+$ is the set of all finite non-empty sequences of elements of $G$. The planner is thus a mapping

$$plan : S \times G \to G^+.$$

At the start of an episode, the planner provides a plan from $s_0$ to the final goal $fg$ : $\mathbf{g}_0 = plan(s_0, fg)$. We denote $\mathbf{g}_t^{(k)}$ the k-th goal of the plan at step $t$. To reduce computational cost, the plan is not recomputed at every step. Instead, it is updated only when the first goal in the current sequence has been reached. Formally:

$$\mathbf{g}_{t+1} = f_{\text{next\_plan}}(s_{t+1}, \mathbf{g}_t, fg) = \begin{cases} \mathbf{g}_t & \text{if } s_{t+1} \notin S_{\mathbf{g}_t^{(1)}} \\ plan(s_{t+1}, fg) & \text{otherwise.} \end{cases} \tag{3}$$

For convenience, we refer to the first goal in the plan as the behavioral goal: $bg_t := \mathbf{g}_t^{(1)}$.

To solve a GCRL problem, a common approach is to combine a planner with a GC-policy. The GC-policy is trained to reach any goal from any starting state. It uses transitions sampled from the replay buffer $(s_t, g, a_t, r_t, s_{t+1})$, where the goal is fixed for the episode, optimizing $\mathcal{M}_{gc}$ (section 2.2), often combined with HER. During environment interaction, however, the GC-policy is conditioned not on the final goal $fg$ but on the behavioral goal provided by the planner: $a_t \sim \pi(s_t, bg_t)$. This ensures the policy is queried only on nearby goals, where it is most accurate, while progressing toward the final goal.

### 2.5 Following a single sequence of low dimensional goals

When the goal and state spaces are identical, each behavioral goal $bg$ specifies a unique target state, so the policy can reach it independently from future goals. In contrast, when the goal space is defined as a low-dimensional projection of the state space, $bg$ can often be reached through multiple distinct state configurations. As shown in (Chenu et al., 2023), although all these configurations are valid to reach the current goal, they may differ in how compatible they are with subsequent goals. As a result, the agent may produce suboptimal trajectories or even fail.

The STIL algorithm (Chenu et al., 2023) is an imitation learning algorithm designed to deal with the case in which chaining two subsequent goals can be an issue. In STIL, the agent learns a behavior from a single demonstration that is split into a unique sequence of goals $\tau_\mathcal{G} = \{g^{(i)}\}_{i \in [1, N_{goals}]}$. The problem is defined as an extended MDP $M_{stil} = \{S_{stil}, A, R_{stil}, P, \gamma, \rho_0, \mathcal{T}, \tau_\mathcal{G}\}$. The agent observes $(s_t, bg_t, i_t) \in S_{stil} = S \times G \times \mathbb{N}$, where $bg_t \in G$ is the current goal the agent is targeting and $i_t$ the index of the goal in sequence $\tau_\mathcal{G}$. When the agent takes an action, it moves to a new state, where $s_{t+1} \sim P(.|s_t, a_t)$ and $(bg_{t+1}, i_{t+1})$ follows:

$$bg_{t+1}, i_{t+1} = f_{stil}(s_t, bg_t, i_t) = \begin{cases} \tau_\mathcal{G}^{(i_t+1)}, i_t + 1 & \text{if } s_{t+1} \in S_{bg_t}, \\ bg_t, i_t & \text{otherwise.} \end{cases}$$

As soon as the agent reaches a goal, a *goal switch* is triggered: it simply switches to the next one. The agent is rewarded each time it reaches its current goal with $R_{stil}(s_t, bg_t, s_{t+1}) = \mathbb{1}[s_{t+1} \in S_{bg_t}]$.
The low-level agent combines SAC and HER and maximizes the expected cumulative reward:

$$\mathbb{E}_{\substack{s_0 \sim \rho_0 \\ bg_{t+1}, i_{t+1} \sim f_{stil}(s_t, bg_t, i_t)}} \left[ \sum_{t=0}^{\infty} \gamma^t R_{stil}(s_t, bg_t, s_{t+1}) \prod_{i=0}^{t} 1 - \mathbb{1}[s_i \in \mathcal{T}(\tau_\mathcal{G}^{(N_{goals})})] \right]. \tag{4}$$

The above formulation has two advantages. First, when the agent aims for its current goal, it also tries to reach it in a configuration that is valid for also reaching the rest of the sequence. Second, since the agent is also conditioned on the current goal index $i_t$, it can differentiate intermediate goals from terminal goals, which helps appropriately handling terminal conditions and using goal relabeling.

## 3 Related Work

Our method enables GC-policies to follow multiple sequences of low-dimensional goals toward any final goal. The focus of our work is the *chaining problem*: ensuring that each intermediate goal is reached in a way that remains compatible with completing the rest of the sequence.

We first situate our method within hierarchical reinforcement learning (HRL), where what we are doing can be seen as focusing on learning a low-level policy given a high-level plan. We then discuss how the chaining problem is addressed by methods that condition GC-policies on full Markov states, on final goals, or on single intermediate goals.

## 3.1 Hierarchical Reinforcement Learning

Hierarchical reinforcement learning provides a framework for decomposing complex tasks into simpler subtasks. A popular approach to HRL is the options framework (Sutton et al., 1999; Bacon et al., 2017) which defines temporally extended actions called options, each consisting of a policy, a termination condition and an initiation set. A high-level policy selects an option, and the option's policy is executed until it terminates. However, the majority of option-based approaches do not explicitly use goals or leverage general goal-conditioned policies. Feudal methods (Dayan & Hinton, 1992; Levy et al., 2019) learn a hierarchy where a high-level policy proposes goals and a low-level GC-policy executes them. In graph-based methods (Eysenbach et al., 2019), a graph is constructed with nodes as states or goals and edges as feasible transitions; a shortest path algorithm then yields a sequence of goals, executed by conditioning the GC-policy on the first goal of the current plan. In our work, we also follow sequences of goals with a low-level GC-policy, as in feudal and graph-based methods, but condition the low-level policy on two goals instead of one. We use a fixed expert planner to abstract high-level planning and compare methods on following sequences of goals assumed to be feasible.

## 3.2 Conditioning GC-policies on Markov states

When intermediate goals are specified as Markov states, the chaining problem is entirely solved by the high-level policy: the low-level policy only needs to reach each intermediate state, without considering the full sequence.

This is the case in HIRO (Nachum et al., 2018) and LEAP (Nasiriany et al., 2019) where the intermediate goals are specified as Markov states, with the GC-policy iteratively conditioned on these states. More recently, Zadem et al. (2024) proposed a two-stage formulation: a first component selects a region of the state space, and a second component generates a sequence of states that progressively guides the policy toward that region. However, learning a low-level policy that reliably reaches Markov states is challenging. As shown in Gehring et al. (2021) the dense reward used in HIRO for the low-level policy is dominated by the $(x, y)$ position, such that the agent often learns to reach $(x, y)$ positions while ignoring other state dimensions (e.g., joint angles). This effectively reduces the problem to conditioning on low-dimensional goals, which can lead to ambiguity: a given goal may be achieved through multiple configurations, introducing chaining issues when composing goals. Moreover, in some practical settings, it is undesirable to specify the goal as a Markov state (e.g., a user may wish to specify only the final position of a robot, not its velocities or sensor states). In our method, we avoid conditioning on Markov states, focusing instead on low-dimensional goals while explicitly addressing the chaining problem.

## 3.3 Conditioning GC-policies on the final Goal

Another line of work trains GC-policies to reach final goals directly, using intermediate goals only as an auxiliary learning signal. Though this sidesteps chaining issues during execution, the auxiliary objectives must be carefully constructed to avoid suboptimal behaviors in the learned low-level GC-policy.

In RIS (Chane-Sane et al., 2021), intermediate goals are part of an auxiliary loss to constrain the GC-policy to perform the same action to reach a distant goal as it would when targeting an intermediate goal along the same path. Since these intermediate goals represent the Markov state, chaining issues are avoided. However, as previously stated, policies conditioned on the Markov states raise specific issues. In GSP (Lo et al., 2024) the authors propose to solve a high-level options SMDP which navigates between way-points. The value function of the high-level SMDP is then used to define a dense, potential-based reward for training a GC-policy that directly targets the final goal. This potential-based reward guides the agent without changing the optimal policy of the original MDP (Ng et al., 1999), ensuring that no chaining problem arises

from approximating the high-level value. Their approach, however, was primarily applied in tabular and discrete-action RL whereas we target continuous states and actions problems.

## 3.4 Myopic GC-policies

Many approaches use GC-policies that depend on a single, low-dimensional goal, e.g. several graph-based (Huang et al., 2019; Lee et al., 2022; Kim et al., 2023), feudal (Levy et al., 2019; Zhang et al., 2022) and options-based (Bagaria et al., 2021) methods. Such policies optimize only for reaching the current goal, without regard for whether the state reached at an intermediate goal is well-suited for completing the rest of the trajectory. This assumption works well in environments like AntMaze (de Lazcano et al., 2024), where the stable body configuration of the ant makes the precise state reached at each goal largely irrelevant. In contrast, for a humanoid agent, reaching the correct spatial position may result in a body configuration that restricts its ability to reliably move farther in all directions. Rather than resolving chaining at the GC-policy level, feudal and graph-based approaches mitigate these issues by adjusting the plan to keep the remaining trajectory feasible given the capabilities of the GC-policy.

**Feudal**: In feudal methods, the high-level policy is rewarded when it proposes a sequence of goals that successfully guides the low-level GC-policy toward the final goal. Extensions typically constrain the high-level policy by requiring goals to be reachable within $k$ steps with the current low-level policy (Levy et al., 2019; Zhang et al., 2022). While this encourages feasible chaining of goals, it also restricts the planner: the planner can only propose sequences that are already compatible with the current low-level skills, potentially excluding otherwise valid solutions.

**Graph-based**: Graph-based methods construct a graph whose nodes correspond to states sampled from the replay buffer, with edges representing the cost of moving from the state stored in one node to the goal representation of another Huang et al. (2019). This formulation implicitly assumes that achieving a goal corresponds to reaching the exact state saved in the target node. In practice, however, an agent may reach the intended goal position through many possible states, some of which can make reaching subsequent goals harder to achieve. To mitigate this mismatch, methods such as Huang et al. (2019) frequently replan, adapting to the trajectory executed by the policy. PIG (Kim et al., 2023) introduces a self-imitation loss, similar to RIS, which encourages the policy to take the same action for a distant goal as for an intermediate goal along the path. Yet, this approach overlooks that the optimal action for an intermediate goal can depend on the subsequent goals. Bonnavaud et al. (2024) report high sample efficiency on AntMaze by combining a pre-trained GC-policy with graph planning. This efficiency is possible under a clearly stated condition: transitions between any two states associated with the same goal are always feasible and costless. This assumption reduces the class of applicable environments and goal abstractions.

**Options skill chaining**: R-DSC (Bagaria et al., 2021) constructs a sequence of options to reach a final goal, where each option corresponds to executing a goal-conditioned policy. To execute a sequence, the first option is selected and its policy is conditioned on a goal that overlaps with the initiation set of the next option. Once the first option reaches its goal, the next option is executed in the same way, and so on, until the final goal is reached. R-DSC updates the initiation set of each option using a classifier that identifies the states from which the current option can reach its goal. It also adjusts the conditioning goal of the policy so that the goal includes the most promising states for subsequent chaining.

However, because each intra-option policy is rewarded simply for reaching a low-dimensional goal before termination, it is optimized to reach any state within the goal set, without regard for whether that state facilitates execution of the subsequent option. Moreover, R-DSC constructs a sequence of options targeting a single final goal, whereas our approach is designed to reach any final goal in the goal space.

A common weakness of all myopic approaches is that the high-level mechanism must adapt to steer the low-level policy along the subset of trajectories that a myopic GC-policy can achieve, which corresponds to a restricted set of trajectories with respect to what a more informed low-level policy could make. In this work, we examine myopic GC-policies in settings that highlight the limitations of focusing exclusively on the current goal. We assume access to an expert planner that proposes sequences of feasible goals, yet myopic policies can still fail or behave sub-optimally. Our approach solves these sequences by explicitly considering future goals when deciding how to reach the current one.

### 3.5 Chaining low dimensional goals

White (2017) introduces a transition-based discount $\gamma(s, a, s')$, so that termination and discounting depend on specific transitions, thereby decomposing a single MDP into subtasks. With a soft discount formulation, the value can be partially propagated across subtask boundaries, enabling the agent to account for rewards that occur beyond the current subtask.

Our work is closer to Chenu et al. (2023), described in Section 2.5, where the agent is trained to imitate a single trajectory composed of a fixed sequence of goals. Our approach extends the work of Chenu et al. (2023) to the general HRL setting, where the agent can produce multiple possible sequences leading to different final goals.

## 4 Methods

In the following sections, we present the problem we address and our method for learning low-level policies that can follow arbitrary sequences of goals. We leverage insights from STIL to construct a specific sequential MDP called xGC-MDP, where the agent prepares to reach future goals while aiming at the current one.

Based on this MDP, we propose two agents: GCP-$final$, which prepares for the full sequence, and GCP-$N$, which prepares only for the next $N$ goals.

### 4.1 Problem statement

We study the problem of solving GC-MDPs, where an agent must learn a policy capable of reaching any specified final goal from any starting state. In our setting, reaching the final goal is framed as following a sequence of intermediate goals. To abstract away high-level planning, we assume access to an expert planner that provides these sequences of intermediate goals.

We focus on the case where goals are low-dimensional relative to the state space, which allows the agent to achieve the same goal through multiple configurations. We study how to design low-level policies following a sequence of low-dimensional goals. In particular, we analyze the chaining issues that arise when GC-agents are conditioned on a single goal and how an agent conditioned on two goals can achieve each goal in a way that remains compatible with the rest of the sequence.

### 4.2 Sequential Goal-Conditioned MDP Formulation

In this section, we propose a sequential MDP framework, where the agent is rewarded for each intermediate goal, integrates goal switches inside the transition function, correctly handles terminal transitions, and is compatible with HER relabeling.

To guide the agent through a sequence of goals, we define a reward function that grants positive feedback not only for reaching the final goal but also for reaching each intermediate goal. In standard $\mathcal{M}_{gc}$, the reward is defined as $R_{gc}(s_t, g, s_{t+1})$ on a goal $g$ that is assumed to be fixed along the episode. In contrast, our setting explicitly models the fact that the next goal the agent must reach evolves over time. Consequently, we define the reward as

$$R_{gc}(s_t, bg_t, s_{t+1}) = \mathbb{1}[s_{t+1} \in S_{bg_t}]$$

where the behavioral goal $bg_t$ denotes the first goal of the current plan $\boldsymbol{g}_t$. Because the reward now depends on $bg_t$, which changes over time, $bg_t$ must be included in the state, and its dynamics must be described in the transition function of the MDP. If the state is defined only as $(s_t, bg_t)$, then transitions take the form

$$(s_t, bg_t) \rightarrow (s_{t+1}, bg_{t+1})$$

where $bg_{t+1}$ depends on the unobserved remaining goal sequence $\boldsymbol{g}_t$. This leads to a Partially Observable Markov Decision Process.

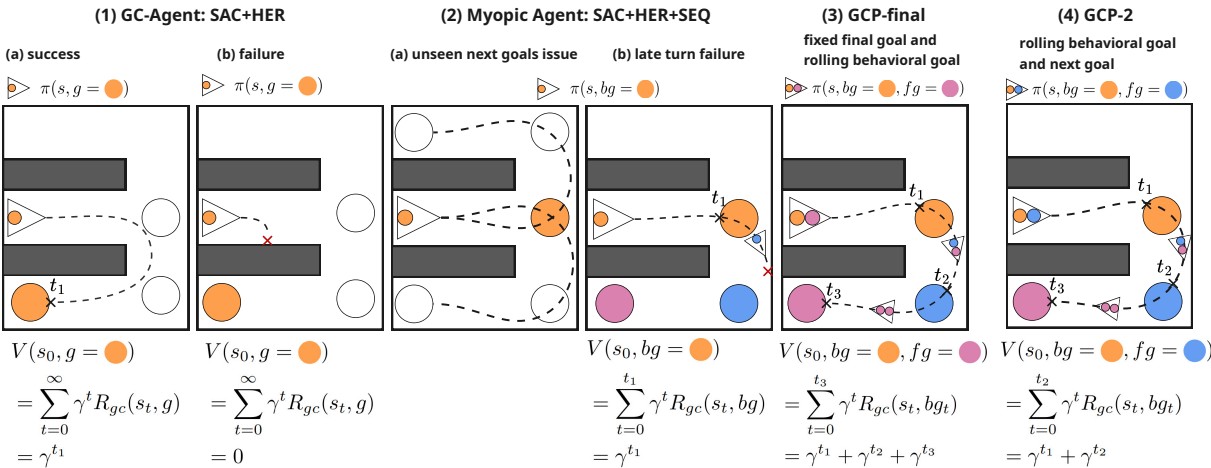

Figure 1: In this didactic example, we compare goal-conditioned formulations in a simple maze. Reaching the final goal is terminal, policies are deterministic. (1) A GC-agent conditioned only on the final goal (a) ignores intermediate goals and can, in principle, reach any goal, but (b) using approximate methods, it often fails on long-horizon goals. (2) A GC-agent using the same loss as in (1), but conditioned on the next goal while interacting with the environment (a) cannot know in advance how to reach its current goal, as it depends on unknown future goals and (b) fails to reach the final goal as the value function ignores the evolution of the behavioral goal. $V(s_0, bg = \bullet)$ sums rewards only until the next goal is reached, since goal reaching is terminal. (3) With a GCP-$final$ agent, the behavioral goal evolves over time both during environment interactions and in the learning objective. The value $V(s_0, bg = \bullet, fg = \bullet)$ includes rewards for all future goals until $t_3$, where the episode terminates upon reaching the final goal. (4) With a GCP-2 agent, both behavioral and final goals evolve along the trajectory. The final goal is assumed fixed when learning, so $V(s_0, bg = \bullet, fg = \bullet)$ sums rewards up to the terminal blue goal at $s_{t_2}$; by being iteratively conditioned on the next two goals, the agent completes the trajectory.

Markovian goal transitions can be preserved by defining an augmented MDP with state $(s_t, \mathbf{g}_t, fg)$, where the agent's policy is conditioned on the complete goal sequence. In this formulation, the next sequence is updated as $\mathbf{g}_{t+1} = f_{\text{next\_plan}}(s_{t+1}, \mathbf{g}_t, fg)$ (see Equation (3)), ensuring that transitions remain Markovian.

It is also possible to preserve the Markov property with a lower-dimensional state by defining the augmented state as $(s_t, bg_t, fg)$. In this formulation, the next goal $bg_{t+1}$ is implicitly determined from the state:

$$bg_{t+1} = f_{\text{next\_goal}}(s_{t+1}, bg_t, fg) = \begin{cases} plan(s_{t+1}, fg)^{(1)} & \text{if } s_{t+1} \in S_{bg_t}, \\ bg_t & \text{otherwise.} \end{cases} \quad (5)$$

In order to properly formalize this process, we propose the following MDP formulation: xGC-MDP = $(S_{gseq}, A, R_{gc}, P, \gamma, \rho_0, \mathcal{T})$. At the beginning of the episode, the planner computes the sequence of intermediate goals up to the final goal $fg$. At each step, the agent observes $(s_t, bg_t, fg) \in S_{gseq} = S \times G \times G$ which is composed of $s_t$, a behavioral goal $bg_t$ that the agent must reach, and the final goal $fg$ that stays fixed during the episode. When the agent takes an action, it moves to a new state $(s_{t+1}, bg_{t+1}, fg)$, where $s_{t+1} \sim P(.|s_t, a_t)$ and $bg_{t+1} = f_{next\_goal}(s_{t+1}, bg_t, fg)$. The reward function is defined as $R_{gc}(s_t, bg_t, s_{t+1}) = \mathbb{1}[s_{t+1} \in S_{bg_t}]$, thus the agent is rewarded for each behavioral goal reached along the trajectory. The objective is to find a policy that maximizes the expected cumulative reward

$$\mathbb{E}_{\substack{s_0 \sim \rho_0 \\ fg \sim P(fg) \\ bg_{t+1} \sim f_{\text{next\_goal}}(s_t, bg_t, fg)}} \left[ \sum_{t=0}^{\infty} \gamma^t R_{gc}(s_t, bg_t, s_{t+1}) \prod_{i=0}^{t} 1 - \mathbb{1}[s_i \in \mathcal{T}(fg)] \right], \quad (6)$$

where $\mathcal{T}(fg)$ is the set of terminal states according to the final goal $fg$. The corresponding temporal difference target is given by:

$$\tilde{Q}(s_t, bg_t, fg, a) = R_{gc}(s_t, bg_t, s_{t+1}) + \gamma \mathbb{1}[s_{t+1} \notin \mathcal{T}(fg)] \max_a Q_\theta^\pi(s_{t+1}, bg_{t+1}, fg, a).$$

As illustrated in Figure 1, the advantages of this formulation are the following. First, because the agent receives a reward for each achieved goal, the optimal policy is the one that completes the entire sequence. Second, it enables proper handling of terminal states, as it can differentiate intermediate and final goals. With the problem framed as an MDP, we can directly use SAC to learn the policy.

**Hindsight Relabeling**: Since the agent is conditioned on two goals, each of them can be relabeled.

Given a sample $(s_t, bg_t, fg, s_{t+1}, bg_{t+1})$, we can relabel $bg_t \leftarrow \phi(s_{k_1})$ and $fg \leftarrow \phi(s_{k_2})$ where $k_1 = \min(t_1, t_2)$ and $k_2 = \max(t_1, t_2)$, with $t_1, t_2$ sampled independently and uniformly from $(t, t_{\max})$. Since xGC-MDP integrates successive goals in the transition dynamics, once the current goal is reached, the next goal must be updated to be the next one the planner would have selected to reach the final goal. When $bg_t$ and $fg$ are relabeled, $bg_{t+1}$ is updated according to Equation (5). Skipping this step would cause the agent to learn from transitions between goals that violate the transition function. The pseudocode is provided in Appendix A.1.

### 4.3 Generalizing from final goal to N-th Goal Conditioning

To solve a GCRL problem with access to a planner, we first propose a method that directly optimizes the entire sequence of goals using the xGC-MDP. We then introduce variants that focus on a limited horizon of future goals.

We denote GCP-$final$ the method that optimizes xGC-MDP using SAC and leverages our modified version of HER. The GCP-$final$ policy is conditioned on the state $(s_t, bg_t, fg)$, which specifies both the current intermediate goal $bg$ and the final goal $fg$. In order to maximize the objective function, the agent should reach $bg$ in a configuration from which it can achieve the rest of the sequence.

In practice, achieving $bg$ in a way compatible with the full sequence may depend primarily on the first few upcoming goals rather than all goals up to $fg$. For example, in a car navigation task, if the next three goals cover an entire turn, at each moment the car can plan for just these three goals. By always preparing for a short horizon of upcoming goals, it can handle a whole sequence of multiple turns without needing to plan for the entire route, similarly to how Model Predictive Control optimizes over a short horizon at each step rather than the full trajectory.

To examine this hypothesis, we introduce a variant, denoted GCP-$N$. This variant employs the same learning algorithm as GCP-$final$ but differs in its action selection strategy during environment interaction. Rather than conditioning on the final goal, the policy is conditioned on the N-th goal in the sequence. More formally, the agent samples actions as

$$a_t \sim \pi(s_t, bg_t, \mathbf{g}_t^{(min(N, len(\mathbf{g}_t)))}),$$

where $\mathbf{g}_t$ is updated with Equation (3), and $len(\mathbf{g}_t)$ corresponds to the number of goals in the sequence $\mathbf{g}_t$.

Given that the GCP-$N$ policy is learned on xGC-MDP, it selects an action that optimizes reaching all intermediate goals up to the $N$-th one. For a GCP-$N$ agent to achieve good performance, it must (1) learn a policy that can always reach the next $N$ goals, and (2) operate in a problem setting where optimizing only the next $N$ goals is sufficient. Failure to satisfy (2) can occur in two ways: the agent may reach an intermediate goal in a configuration that is not compatible with reaching the final goal leading to failure, or it may complete the full sequence but with suboptimal behaviors. In contrast, GCP-$final$ does not require property (2), since it directly optimizes the full sequence. However, for GCP-$final$, property (1) may be harder to achieve. Shorter horizons improve value function estimation (Park et al., 2023), and during training the agent is more likely to observe transitions involving nearby goals than distant ones.

The pseudocode for GCP-$final$ and GCP-$N$ is provided in Appendix A.1.

## 5 Experiments

Our experimental study is designed to evaluate three types of agents: (i) SAC+HER, an agent conditioned on the final goal without leveraging planning, (ii) SAC+HER+SEQ: an agent conditioned iteratively on the successive goals provided by the expert planner, corresponding to the formulation adopted by most feudal and graph-based approaches (see Section 2.4), and (iii) GCP-$N$: our agents conditioned on both the first and $N$-th goals of the plan, as described in Section 4.3.

All these agents rely on SAC+HER with the same hyper-parameters (see Appendix A.3), conclusions are therefore only drawn under these configurations, without specific fine-tuning for each agent.

We evaluate them in four environments with various goal chaining difficulties and various terminal conditions to highlight the advantages and limitations of the different approaches. All agents are trained on goals sampled uniformly from the goal space. Evaluations are performed on a fixed set of challenging (start state, goal) pairs specified for each environment in Figure 2.

Evaluation metrics are reported using both figures and tables. Figures show the mean performance over the evaluation set during training, including success rate, number of rollout steps required to reach the goal, or cumulative reward. The associated tables report, for the same runs, the final success rate and the number of training time steps required to reach a 99% success rate.

### 5.1 Environments

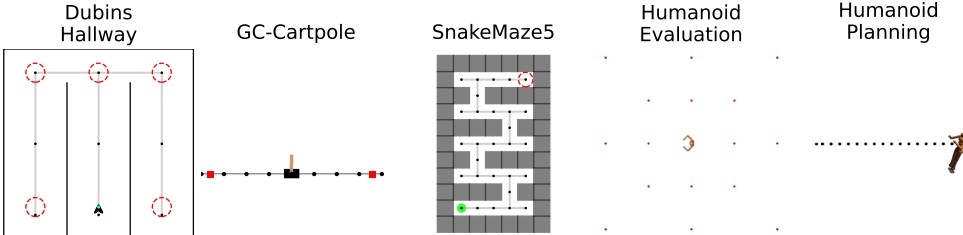

Figure 2: Each environment shows the evaluation goals together with the expert graph used for planning. In *Dubins Hallway*, the agent is evaluated on five goals, starting in the main corridor and aiming for each goal shown as a red circle. In *GC-Cartpole*, the agent starts at the bottom center and is evaluated on two goals, shown as red rectangles. In *SnakeMaze5*, the agent starts in the green area and is evaluated only on the hardest goal, shown as a red circle. In *GC-Humanoid-Walk*, the agent is evaluated on all 16 goals, shown as red circles. The starting position is always at the center of the arena, facing the current evaluation goal. In this environment, the planner does not use a graph. Instead, it outputs uniformly spaced points between the agent and the final goal.

We evaluate the methods in four environments, each highlighting different goal-chaining challenges. In Dubins Hallway, ignoring future goals can lead to collisions with walls, resulting in failure. PointMaze and Cartpole illustrate how terminal and non-terminal conditions, respectively, can affect performance in sequences of goal following. Finally, GC-Humanoid-Walk tests scalability to high-dimensional action and state spaces, where each intermediate goal must be reached in a sufficiently balanced configuration to allow continued locomotion toward the final goal.

**Dubins Hallway** is a navigation task in which the agent controls a car in a 2D maze. The state $s = \{x, y, cos(\theta), sin(\theta)\}$ includes the position and orientation of the agent. Each goal $g$ is defined as a position $(x, y)$ and is considered reached if the agent is within a ball of radius 0.1 centered at that position. The agent moves forward at a fixed speed at each step, the action controls the variation of the orientation $\dot{\theta}$ for the agent. During a *training* episode, both the initial position of the agent and the goal are randomly sampled within the maze boundaries. During an *evaluation* episode, the agent is tested on a fixed start state and multiple goal locations illustrated in Figure 2. If the agent hits a wall, it stays stuck until the episode ends. The state is only terminal when the agent reaches the goal. A key feature of this environment is that, when

the agent is in the central hallway, it should not prepare for the next goals in the same way depending on whether the final goal is on the left- or right-hand side. In contrast, the velocity of the agent being constant, the time-to-goal does not vary much in this environment, so we do not present results on this aspect.

In **GC-Cartpole**, the agent must reach a given position while balancing a pole. The state $s = (\dot{x}, \theta, \dot{\theta})$ contains velocity $\dot{x}$, angle $\theta$, and angular velocity $\dot{\theta}$. As explained in more detail in Appendix A.2, the agent's goal is the difference between its current position $x$ and a fixed target position $x_{dg}$. The goal is reached once $||x - x_{dg}|| < 0.05$. Actions are continuous values in $[-1, 1]$ proportional to the force applied to the cart. During a *training* episode, the agent targets a random uniform goal in the range $[-5, 5]$ starting from $[0, 0]$. During a *evaluation* episode, the agent is tested on multiple goal locations illustrated in Figure 2. A state is terminal when $\theta$ leaves the $[-0.12°, 0.12°]$ interval. As reaching a goal is not terminal, the agent must learn to reach its target position and stay there while keeping the pole balanced.

In **SnakeMaze5** (de Lazcano et al., 2024), the agent controls a ball navigating in a 2D maze. The state $s = \{x, y, \dot{x}, \dot{y}\}$ includes the position and velocity of the agent. Each goal $g$ is defined as a $(x, y)$ position and is considered reached if the agent is within a ball of radius 0.45 centered at that position. The action represents the linear force exerted on the ball in the x and y directions. In de Lazcano et al. (2024), the agent's velocity was limited to the range [-5,5] m/s. Within this range, the agent could easily take sharp turns at maximum speed, making the task relatively simple. To increase the difficulty, we extended this limit to [-10, 10] m/s[1], requiring the agent to anticipate turns and decelerate in advance. During a *training* episode, both the agent's initial position and the goal are randomly sampled within the maze boundaries. During a *evaluation* episode, the agent is tested on multiple goal locations illustrated in Figure 2. The state is only terminal when the agent reaches the goal.

In **GC-Humanoid-Walk**, we adapted the original humanoid-V5 environment (Towers et al., 2024) to the goal-conditioned case. The agent must learn to move toward target positions in an empty arena. The state $s \in \mathcal{R}^{348}$ includes joint positions, velocities, inertial properties, center-of-mass velocities, actuator forces, and external forces on body parts. As in *GC-Cartpole*, goals are not given as $(x, y)$ absolute positions, but as relative differences between the current agent position and a target position. A goal is reached if $||(x_{torso}, y_{torso}) - (x_{goal}, y_{goal}) < 0.1||$. When the z-coordinate of the torso (the height) is not in the closed interval $[1, 2]$ the episode terminates and a negative reward of -10 is given. During each *training* episode, we sample a random uniform goal in the square $(x, y) \in [-5, 5] \times [-5, 5]$. During a *evaluation* episode, the agent is tested on multiple goal locations illustrated in Figure 2. The humanoid agent is always initialized at the center of the environment, oriented to face the sampled goal. In this task, reaching the final goal does not terminate the episode; instead, the agent must learn to reach the target position and remain in equilibrium indefinitely.

**Planning**: In the *GC-Humanoid-Walk* environment, the planner sets intermediate goals evenly spaced between the agent and the final goal. In all other environments, the planner computes the shortest path using the expert graphs shown in Figure 2.

## 5.2 Conditioning on increasingly distant goals

We study the impact of the index of the second goal $N$ used to condition the agent policy (Section 4.3) where $N$ is between 2 and the index of the final goal[2].

Learning curves are shown in Figure 3 while Table 1 reports the mean final success rate and the mean timestep at which a 99% success rate was reached. Learning a policy with smaller values of $N$ yields comparable performance with fewer environment interactions in *SnakeMaze5* and *Dubins Hallway* and improved performance in *GC-Cartpole* and *GC-Humanoid-Walk*. In particular, the best performances or sample efficiency are observed for $N = 2$, corresponding to the setting in which the agent is conditioned on the next two upcoming goals, while it tends to be the lowest for $N = $ final.

---

[1]The upper bound was determined empirically; beyond this value, collisions become unstable and the agent can occasionally pass through maze walls.

[2]GCP-1 would correspond to conditioning the agent twice on the next goal, which is similar to SAC+HER+SEQ, studied in Section 5.3 and Section 5.4.

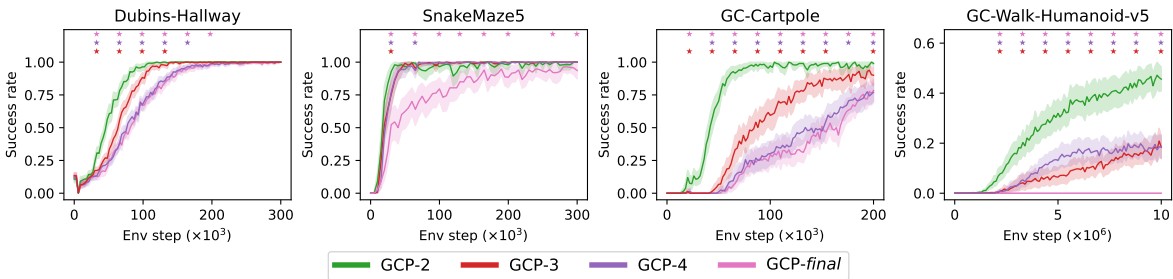

Figure 3: Performance of GCP-$N$ as a function of $N$, the number of upcoming goals included in the planning horizon. Smaller $N$ improves sample efficiency or overall performance (statistical tests supporting this claim are reported in Table 1). Performance is reported as the mean success rate on the evaluation goal set defined in Figure 2, Each evaluation is performed 10 times, with results reported as the mean and 95% confidence interval (computed with bootstrapping) over 50 seeds. Stars denote statistically significant differences (Welch T-test, p-value $< 0.05$) between GCP-2 and the algorithm of the corresponding color.

Table 1: Mean $\pm$ standard deviation of the final success rate and the first timestep at which a 99% success rate (SR) was reached over 50 seeds. $^\star$ indicates a statistically significant difference from GCP-2 (Welch T-test, p-value $< 0.05$). Blank entries indicate that there was at least one run in which 99% SR was not reached.

| Env | Metric | GCP-2 | GCP-3 | GCP-4 | GCP-$final$ |
|---|---|---|---|---|---|
| Dubins | Final SR | $1.0 \pm 0$ | $1.0 \pm 0$ | $1.0 \pm 0$ | $1.0 \pm 0$ |
| | 99% SR at | $63.4$k $\pm 11.9$k | $86.3$k $\pm 14.8$k $^\star$ | $120.3$k $\pm 30.2$k $^\star$ | $124.7$k $\pm 25.0$k $^\star$ |
| SnakeMaze5 | Final SR | $1.0 \pm 0$ | $1.0 \pm 0$ | $1.0 \pm 0$ | $0.94 \pm 0.22$ $^\star$ |
| | 99% SR at | $20.7$k $\pm 4.9$k | $25.6$k $\pm 6.6$k $^\star$ | $26.9$k $\pm 6.1$k $^\star$ | $49.7$k $\pm 30.0$k $^\star$ |
| Cartpole | Final SR | $0.99 \pm 0.07$ | $0.90 \pm 0.25$ $^\star$ | $0.77 \pm 0.26$ $^\star$ | $0.78 \pm 0.32$ $^\star$ |
| | 99% SR at | $53.0$k $\pm 13.4$k | $-$ | $-$ | $-$ |
| Humanoid | Final SR | $0.46 \pm 0.17$ | $0.18 \pm 0.16$ $^\star$ | $0.18 \pm 0.17$ $^\star$ | $0.00 \pm 0^\star$ |
| | 99% SR at | $-$ | $-$ | $-$ | $-$ |

One challenge for GCP-$final$ is that the second goal used to condition the policy can be any goal in the goal space. This requires the policy to generalize over a much wider range of goals, which increases learning difficulty. By contrast, in GCP-2, the second goal corresponds to a restricted subset of goals (the second goals of any plan), rather than the entire goal space. Moreover, when HER is applied, the final goal is often replaced by goals that were actually achieved in successful trajectories. On average, these substituted goals are closer to the current state of the agent, so training transitions contain fewer examples with distant goals, which are necessary for GCP-$final$ but not GCP-2.

The main advantage of GCP-$final$ compared to GCP-2 is that GCP-$final$ can prepare not only for the next two goals but also for all subsequent ones. This benefit depends on the ability to back-propagate rewards from distant goals through the value function. If this propagation fails, due to limited generalization, bias from HER, or the difficulty of long-horizon credit assignment, the policy cannot use the additional information. To examine this point, we analyze the value functions learned by each GCP-$N$ variant.

For each GCP-$N$ variant, we assess whether the learned value accurately estimates the expected return and correctly retro-propagates rewards from the first $N$ goals.

Formally, we compare the learned value $V_\theta^\pi(s_t, bg_t, \mathbf{g}_t^{(N)})$ with the theoretical value $V^\pi(s_t, bg_t, \mathbf{g}_t^{(N)})$ of the current policy. Since SAC does not directly compute $V_\theta^\pi$, but only $Q_\theta^\pi$, we estimate:

$$V_\theta^\pi(s_t, bg_t, \mathbf{g}_t^{(N)}) = Q_\theta^\pi(s_t, bg_t, \mathbf{g}_t^{(N)}, \pi(s_t, bg_t, \mathbf{g}_t^{(N)})).$$

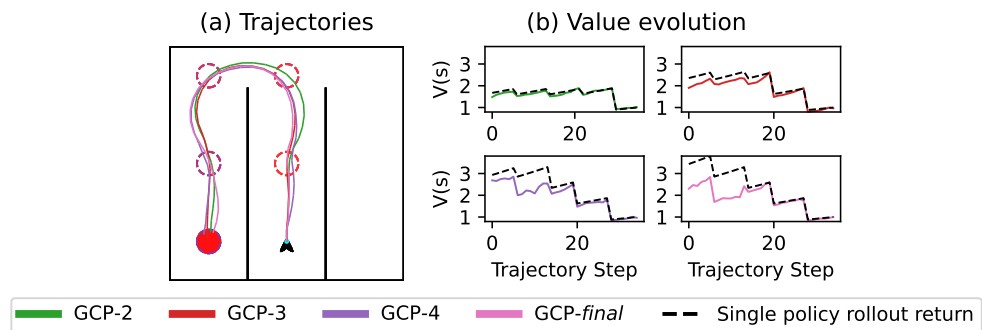

Figure 4: Qualitative analysis of GCP-$N$ agents in *Dubins Hallway*. (a) Set of trajectories of trained agents (b) Value function and single policy rollout return over episodes matching trajectories in (a). GCP-2 shows the best sample efficiency. As $N$ increases, the value estimates deviate more from the single policy rollout return.

In deterministic environments, with deterministic policies, a single policy rollout return and $V^\pi$ coincide, as it is the deterministic counterpart of the unbiased Monte Carlo value estimate Sutton & Barto (1998)[3]. This policy rollout return is computed as

$$MC(t, N) = \sum_{k=0}^{T|s_T \in \mathbf{g}_t^{(N)}} \gamma^k r_{t+k+1}.$$

Figure 4(b) shows that the discrepancy between $V$ and $MC$ increases with $N$. This result suggests that the critic trained with SAC has only propagated the value from the first few goals of the sequence. According to this interpretation, since the agent only prepares for reaching the first few intermediate goals and mostly ignores the influence of more distant goals, conditioning the policy on the last goals in the plan does not provide a practical benefit.

Other factors might come into play, such as the fact that the agent accumulates fewer experience in reaching goals that are more distant. We further investigate the question in Appendix A.7.

The key takeaway of these experiments is that GCP-2 consistently achieves equal or superior performance by preparing for the first two goals. Thus, we focus on GCP-2 in subsequent experiments.

## 5.3 Comparison to baselines

In this section, we compare success rates of our agent conditioned on two goals with two approaches in which the policy is conditioned on a single goal.

As Figure 5 and Table 2 show, SAC+HER consistently achieves high success rates in *Dubins Hallway*, where the time horizon is short (40 timesteps), and in *GC-Cartpole*, where the dynamics is simple. However, in environments such as *SnakeMaze5* or *GC-Humanoid-Walk*, SAC+HER catastrophically fails, never reaching goals that are distant from the initial position.

In contrast, SAC+HER+SEQ shows the best sample efficiency in *GC-Cartpole* and *SnakeMaze5*. However, the resulting behaviors are not always optimal, as discussed in Section 5.4. In *Dubins Hallway*, insufficient anticipation of future goals can cause colliding with walls. In *GC-Humanoid-Walk*, it can result in a loss of balance (Figure 6). This illustrates that this method fails in environments where preparing for future goals is critical.

Our method, GCP-2, consistently achieves the highest or comparable success rates in all environments.

---

[3]SAC employs a stochastic policy; however, stochasticity vanishes at the end of training across repeated rollouts.

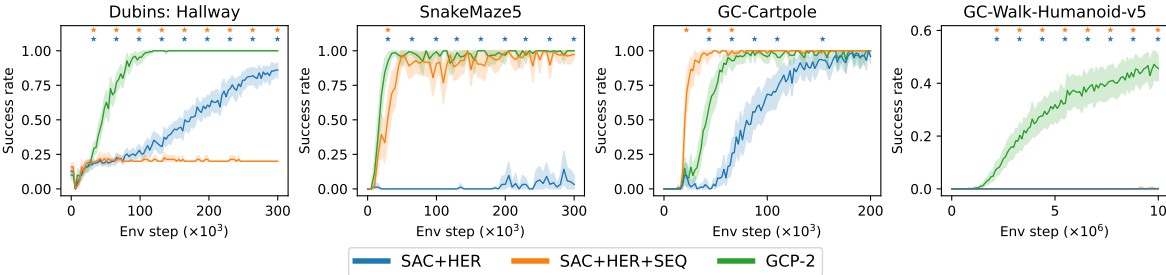

Figure 5: Success rates. SAC+HER+SEQ achieves the best sample efficiency in *GC-Cartpole*, but fails in *GC-Humanoid-Walk* and *Dubins Hallway*. In contrast, GCP-2 attains similar or better success rates across all environments (statistical tests supporting this claim are reported in Table 2). Results are reported as the mean success rate on the evaluation goal set defined in Figure 2. Each evaluation is performed 10 times. Shaded regions indicate the 95% confidence interval (computed with bootstrapping) over 50 seeds. Stars denote statistically significant differences (Welch T-test, p-value < 0.05) between GCP-2 and the algorithm of the corresponding color.

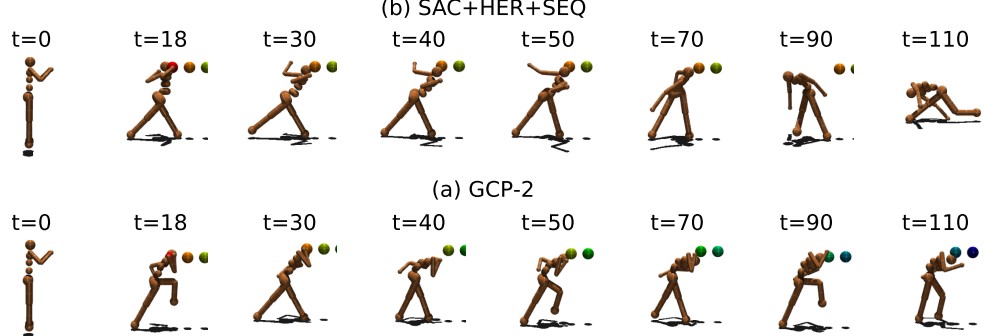

Figure 6: Trajectories of fully trained *GC-Humanoid-Walk* agents. (a) SAC+HER+SEQ reaches the first goal in a way that prevents completing the full sequence (e.g., taking an overly wide step). (b) GCP-*N* reaches the first goal in a configuration compatible with the second, enabling the chaining of multiple goals.

Table 2: Mean ± standard deviation of the final success rate and the first timestep at which a 99% success rate (SR) was reached over 50 seeds. $^\star$ indicates a statistically significant difference from GCP-2 (Welch T-test, p-value < 0.05). Blank entries indicate that there was at least one run in which 99% SR was not reached.

| Env | Metric | GCP-2 | SAC+HER | SAC+HER+SEQ |
|---|---|---|---|---|
| Dubins | Final SR | $1.0 \pm 0$ | $0.86 \pm 0.16 ^\star$ | $0.20 \pm 0^\star$ |
| | 99% SR at | $63.4\text{k} \pm 11.9\text{k}$ | – | – |
| SnakeMaze5 | Final SR | $1.0 \pm 0$ | $0.02 \pm 0.14 ^\star$ | $0.98 \pm 0.07$ |
| | 99% SR at | $20.7\text{k} \pm 4.9\text{k}$ | – | $31.0\text{k} \pm 9.6\text{k} ^\star$ |
| Cartpole | Final SR | $0.99 \pm 0.07$ | $0.93 \pm 0.17 ^\star$ | $0.99 \pm 0.07$ |
| | 99% SR at | $53.0\text{k} \pm 13.4\text{k}$ | $100.3\text{k} \pm 30.6\text{k} ^\star$ | $23.6\text{k} \pm 5.0\text{k} ^\star$ |
| Humanoid | Final SR | $0.46 \pm 0.17$ | $0.00 \pm 0^\star$ | $0.00 \pm 0.00 ^\star$ |
| | 99% SR at | – | – | – |

## 5.4 Effects of terminal conditions

Terminal conditions play an important role in GC-RL and can change the nature of the task. We consider two cases. (i) In environments with terminal goals, reaching the goal terminates the episode. In that case, the optimal behavior is to reach the goal as quickly as possible. (ii) When goals are non-terminal, reaching them does not terminate the episode. In that case, the optimal behavior is to reach the goal and remain in equilibrium, since this yields an infinite sum of discounted positive rewards.

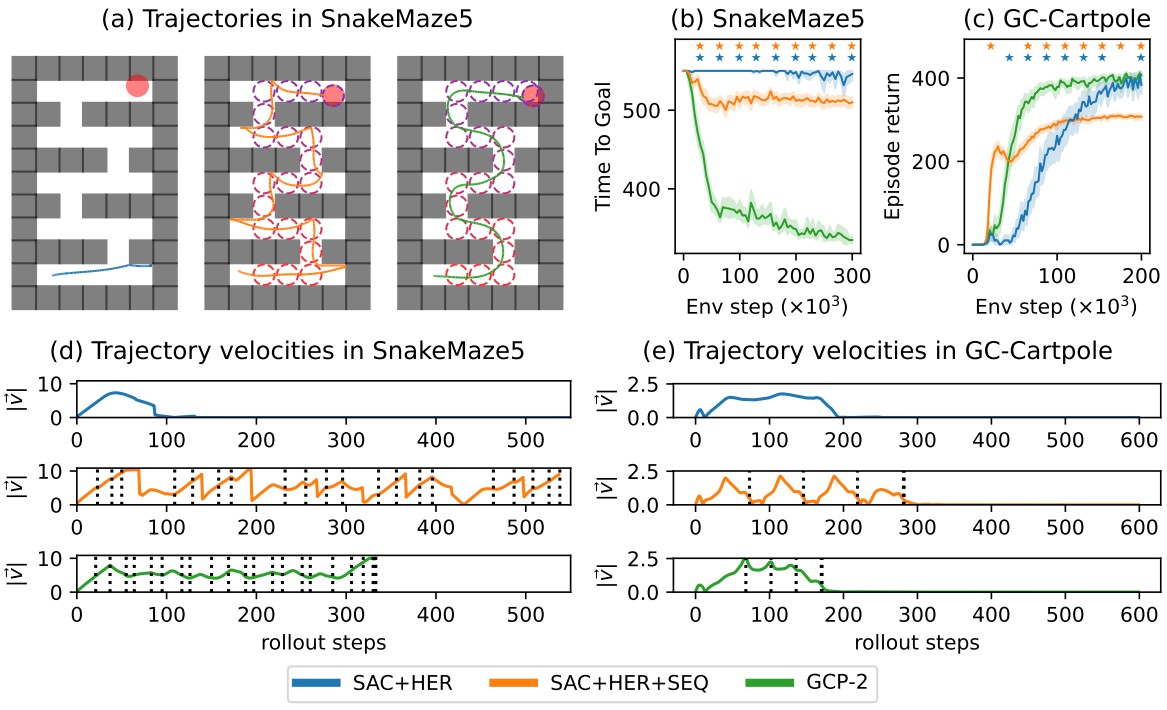

Figure 7: Analysis of behaviors in *SnakeMaze5* and *GC-Cartpole*, whose goals are terminal and non-terminal respectively. The evaluation goal sets are defined in Figure 2. (a) Some trajectories generated by agents trained in *SnakeMaze5*. (b) Mean time to goal in *SnakeMaze5*. (c) Mean episode return in *GC-Cartpole*. (d, e) Velocities of trajectories generated by agents trained in *SnakeMaze5* and *GC-Cartpole*. For (b,c), results are reported as the mean and 95% confidence interval (computed with bootstrapping) over 50 seeds. Stars denote statistically significant differences (Welch T-test, p-value < 0.05) between GCP-2 and the algorithm of the corresponding color. SAC+HER+SEQ converges to a suboptimal behavior compared to GCP-2 in both environments (statistical tests supporting this claim are reported in Table 2). GCP-2 manages intermediate goals with a more stable velocity.

Table 3: Mean ± standard deviation over 50 seeds. For SnakeMaze5, we report final time to goal (lower is better). For Cartpole, we report episode return (higher is better). ⋆ indicates a statistically significant difference from GCP-2 (Welch's t-test, $p < 0.05$).

| Env | Metric | GCP-2 | SAC+HER | SAC+HER+SEQ |
|---|---|---|---|---|
| SnakeMaze5 | time to goal ↓ | 334.9 ± 16.7 | 546.1 ± 27.7⋆ | 510.0 ± 23.2⋆ |
| Cartpole | return ↑ | 406.8 ± 34.4 | 383.1 ± 72.2⋆ | 307.1 ± 25.0⋆ |

Figures 7(d, e) illustrate the velocities of agents in *SnakeMaze5* and *GC-Cartpole*. In these figures, black vertical dotted lines correspond to time steps where the agent reaches an intermediate goal.

In *SnakeMaze5*, SAC+HER+SEQ reaches goals at high speed (Figure 7 (d)). However, this behavior makes it difficult to turn, resulting in bouncing off walls (Figure 7 (a)). As shown in Figure 7(b) and Table 3, the time the agent needs to reach the final goal is longer than with GCP-2 because the SAC+HER+SEQ agent does not distinguish intermediate from final goals.

In *GC-Cartpole* (Figure 7(e)), the SAC+HER+SEQ agent decelerates before each goal to ensure it can stabilize there. This behavior results in overall slower movement compared to other methods (Figure 7(c),Table 3).

In contrast, SAC+HER, which ignores intermediate goals, obtains a high return in *GC-Cartpole* but frequently fails in *SnakeMaze5*, as the agent struggles to reach distant goals. Finally, GCP-2 performs equally or better in both environments, with a more constant velocity in *SnakeMaze5* and *GC-Cartpole* (Figure 7(d, e)). Conditioning on the two upcoming goals allows the agent to anticipate future turns in *SnakeMaze5* and to cross intermediate goals in *GC-Cartpole* with only minimal deceleration.

## 5.5 HER ablation

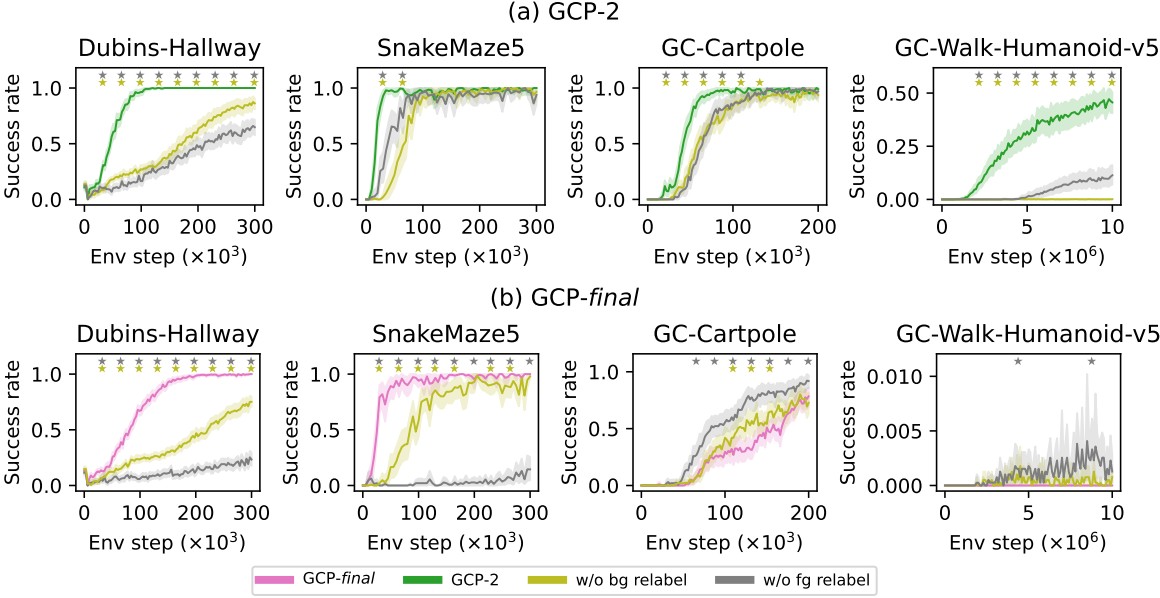

Figure 8: HER ablation, showing that success rate is generally higher with HER than without for GCP-2 (top row) and GCP-*final* (bottom row), statistical tests supporting this claim are reported in Table 4. Results are reported as the mean success rate on the evaluation goal set defined in Figure 2. Each evaluation is performed 10 times, with results reported as the mean and 95% confidence interval (computed with bootstrapping) over 50 seeds. Stars denote statistically significant differences (Welch T-test, p-value $< 0.05$) between GCP-2 (top row) or GCP-*final* (bottom row) and the algorithm of the corresponding color.

We evaluate the role of HER by examining the effect of relabeling both the behavioral goal and the final goal in the GCP-2 and GCP-*final* settings across all previously considered environments.

Figure 8 and Table 4 show that, in most environments, omitting the relabeling of either the behavioral goal or the final goal leads to a drop of success rate or sample efficiency. In particular, HER proves to be more effective in *GC-Humanoid-Walk*, where relabeling considerably improves the final success rate of the GCP-2 method. An exception is observed in *GC-Cartpole*, where the impact of relabeling is low with the GCP-2 method and results in a small drop of success rate for GCP-*final*.

Table 4: Mean ± standard deviation of the final success rate and the first timestep at which a 99% success rate (SR) was reached over 50 seeds. $\star$ indicates a statistically significant difference (Welch T-test, p-value $< 0.05$) between GCP-2 or GCP-$final$ and their respective ablations. Blank entries indicate that there was at least one run in which 99% SR was not reached.

| Env | Metric | GCP-2 | GCP-2 w/o bg | GCP-2 w/o fg | GCP-$final$ | GCP-$final$ w/o bg | GCP-$final$ w/o fg |
|---|---|---|---|---|---|---|---|
| Dubins | Final SR | $1.0 \pm 0$ | $0.86 \pm 0.19^\star$ | $0.65 \pm 0.25^\star$ | $1.0 \pm 0$ | $0.75 \pm 0.17^\star$ | $0.23 \pm 0.20^\star$ |
| | 99% SR at | $63.4k \pm 11.9k$ | – | – | $124.7k \pm 25.0k$ | – | – |
| SnakeMaze5 | Final SR | $1.0 \pm 0$ | $0.97 \pm 0.19$ | $0.94 \pm 0.21$ | $1.0 \pm 0$ | $0.98 \pm 0.06$ | $0.14 \pm 0.30^\star$ |
| | 99% SR at | $20.7k \pm 4.9k$ | $66.7k \pm 14.3k^\star$ | $46.9k \pm 15.0k^\star$ | $31.2k \pm 6.8k$ | $89.5k \pm 25.2k^\star$ | – |
| Cartpole | Final SR | $0.99 \pm 0.07$ | $0.94 \pm 0.19$ | $0.97 \pm 0.16$ | $0.78 \pm 0.32$ | $0.73 \pm 0.35^\star$ | $0.92 \pm 0.21^\star$ |
| | 99% SR at | $53.0k \pm 13.4k$ | – | $74.9k \pm 22.4k^\star$ | – | – | – |
| Humanoid | Final SR | $0.46 \pm 0.17$ | $0.00 \pm 0.00^\star$ | $0.11 \pm 0.16^\star$ | $0.00 \pm 0$ | $0.00 \pm 0.00^\star$ | $0.00 \pm 0.00^\star$ |
| | 99% SR at | – | – | – | – | – | – |

## 6 Conclusion

In this paper, we have studied the problem of following diverse sequences of low-dimensional goals from a given plan. We showed that strategies relying only on the immediate next goal can lead to failure cases. To address this, we proposed a new MDP formulation, in which we formalize goal transitions and terminal states to ensure that the objective is to reach all goals in the sequence. Within this framework, we proposed the GCP-$N$ method, where the agent prepares to reach up to $N$ goals. Through navigation, pole-balancing and locomotion experiments, we have shown that in most environments, agents trained with GCP-2 are more sample-efficient, or achieve higher performance than non sequential, myopic and GCP-$N$ agents with a larger $N$.

A first limitation of our approach is that GCP-$N$ variants consider only the first $N$ goals in the sequence. Theoretically, this means that the optimal policy of the corresponding problem can be suboptimal compared to directly targeting the final goal or optimizing over the whole sequence, such as GCP-$final$. However, in practice, particularly in the challenging *GC-Humanoid-Walk* environment, learned policies are far from their theoretical optimum, and we showed that restricting optimization to the first few goals facilitates learning and produces empirically strong performance. An interesting direction for future work would be to exhibit environments where using a larger $N$ is mandatory and to make $N$ adaptive to the complexity of the sequence, so that the agent prepares further ahead only when necessary.

In this work, we used a fixed oracle planner, which allowed us to focus on analyzing the low-level learning of GC-policies for goal sequences. Our method would still work with a learned planner, provided that the planner is trained first and provides good enough intermediate goals, before learning the GC-policy. We leave the case where intermediate goals are misleading for future work. A second limitation of our approach arises when GC-policies and the planner are learned in parallel.

If the planner evolves over time, the low-level MDP can become non-stationary, depending on how the behavioral goal dynamics are defined. When the behavioral goal is fixed throughout the episode (as in SAC+HER+SEQ), the dynamics remain stationary. In contrast, if the behavioral goal switches to the next goal provided by the planner (as in GCP-$N$), each change in the planner modifies the low-level MDP and can introduce learning instabilities. However, the less the agent prepares for future goals, the fewer planner-dependent behavioral goal transitions. An extreme case occurs when the agent is conditioned on the next two goals of a plan: in this case, the evolution of behavioral goals becomes planner-independent. In such cases, a GCP-$N$ agent conditioned on $\pi(..., bg = A, fg = B)$ can only switch to $\pi(..., bg = B, fg = B)$ once the goal $A$ has been reached. Consequently, GCP-$N$ with smaller $N$ is likely to be more stable. Exploring strategies for the simultaneous learning of both components is left for future work.

**Broader Impact Statement**

As any work in reinforcement learning, our work may be relevant for applications in robotics, navigation, or other sequential decision-making tasks. One must be aware that policies learned through reinforcement learning can behave unpredictably in novel situations, which can be problematic in safety-critical or high-stakes contexts, highlighting the importance of careful evaluation and human oversight. Most importantly, these potential issues are not specific to our work.

**Reproducibility**

The code for this study is publicly available at `https://github.com/olivier-serris/implementation-tale-of-two-goals`.

**Acknowledgments**

This was performed using HPC resources from GENCI-IDRIS (Grant 2022-AD011015961). It also received funding from the European Commission's Horizon Europe Framework Program under grant agreement No 101070381 (PILLAR-robots project). The authors also thank Alexandre Chenu for helpful discussions.

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

# A   Appendix

## A.1   Pseudo-code

Algorithm 1 provides the pseudocode for the main loop, illustrating how GCP-$N$ and GCP-$final$ operate within the environment using a planner. Additionally, Algorithm 2 outlines the relabeling mechanism.

---
**Algorithm 1** Main Loop

---
**Require:** $Q$, $\pi$, planner, env, N, $\phi$
  $RB \leftarrow []$
  **for** $N = 1 : N_{episodes}$ **do**
    $s, env_g \leftarrow env.reset()$
    $path \leftarrow [g^{(1)}, g^{(2)}, ..., env_g] \leftarrow planner.path(s, env_g)$
    $bg \leftarrow path[0]$
    **if** $N \neq None$ **then**
      $fg \leftarrow path[N-1]$
    **else**
      $fg \leftarrow env_g$
    **end if**
    $done \leftarrow false$
    **while** not done **do**
      $a \leftarrow \pi(a|s, bg, fg)$
      $s', r, term, trunc \leftarrow \text{env.step}(a)$
      **if** $s' \in bg$ **then**
        $path \leftarrow [g^{(1)}, g^{(2)}, ..., env_g] \leftarrow planner.path(s, env_g)$
        $bg' \leftarrow path[0]$
        **if** $N \neq None$ **then**
          $fg \leftarrow path[N-1]$
        **end if**
      **end if**
      $RB \leftarrow RB + [(s, a, r, bg, fg, s', bg', term)]$
      $s, bg = s', bg'$
      $Q, \pi \leftarrow \text{Learn\_Step}(Q, \pi, RB, \text{planner}, \text{env.terminal\_func}, \phi)$
      $done \leftarrow term \vee trunc$
    **end while**
  **end for**

---

## A.2   Absolute and relative goals

Absolute goals are goals that correspond to absolute positions in the goal space, while relative goals are goals whose positions are relative to the agent. The relative goal formulation is better suited in *GC-Cartpole.*

---

**Algorithm 2** Learn_Step

---

**Require:** $Q$, $\pi$, RB, planner, terminal_func

    $\tau \leftarrow$ Sample trajectory from RB

    $s_t, a_t, bg_t, r_t, fg, s_{t+1}, bg_{t+1}, term_t \leftarrow$ Sample transition from $\tau$

    $relabel \leftarrow (\text{random}(0.0, 1.0) < 0.8)$

    **if** $relabel$ **then**

        $states \leftarrow [s_0, s_1, ..., s_{t_{max}}] \leftarrow$ Extract all states from the trajectory $\tau$

        $t_1 \sim random(t, t_{max})$

        $t_2 \sim random(t, t_{max})$

        $k_1 \leftarrow min(t_1, t_2)$

        $k_2 \leftarrow max(t_1, t_2)$

        $bg_t, fg \leftarrow \phi(states[k_1]), \phi(states[k_2])$

    **end if**

    **if** $s_{t+1} \in bg_t$ **then**

        $bg_{t+1} \leftarrow \text{planner.path}(s_t, fg)[0]$

    **else**

        $bg_{t+1} \leftarrow bg_t$

    **end if**

    $r_t \leftarrow 1$ **if** $s_{t+1} \in bg_t$ **else** $0$

    $term_t \leftarrow \text{terminal\_func}(s_t, a_t, s_{t+1}, fg)$

    $\text{transition} = s_t, a_t, bg_t, r_t, s_{t+1}, bg_{t+1}, fg, term_t$

    $\pi, Q \leftarrow \text{SAC\_Update}(\text{transition}, \pi, Q)$

    **return** $\pi, Q$

---

Indeed, with an absolute goal formulation, going from $x = -2$ to $x = 0$ and from $x = 8$ to $x = 10$ are two different things, whereas with a relative goal formulation, the agent must only learn how to go to $x_{rel} = 2$ to solve both cases at once. Similarly, in *GC-Humanoid-Walk*, the goal is defined independently of the agent's absolute location: the agent is required to reach a target $(x, y)$ position specified in relative coordinates.

## A.3 Hyper-parameters

The hyper-parameters used for each environment are presented in Table 5. All baselines and our methods share the same hyper-parameters, taken from the original SAC paper Haarnoja et al. (2018), except for SAC+HER directly targeting the final goal, where we use a higher discount factor 0.995 for long horizon tasks (*SnakeMaze5*, *GC-Cartpole*, *GC-Humanoid-Walk*). We only increased the neural network size for the most complex tasks, as in Chenu et al. (2023).

Table 5: Hyper-parameters used for SAC

| Hyper-parameters | *Dubins Hallway* | *SnakeMaze5* | *GC-Cartpole* | *GC-Humanoid-Walk* |
|---|---|---|---|---|
| Random actions | $5K$ | $5K$ | $5K$ | $50k$ |
| Critic hidden size | $[256, 256]$ | $[512, 512]$ | $[256, 256]$ | $[512, 512, 512]$ |
| Policy hidden size | $[256, 256]$ | $[512, 512]$ | $[256, 256]$ | $[512, 512, 512]$ |
| Activation functions | ReLU | ReLU | ReLU | ReLU |
| Batch size | 256 | 256 | 256 | 256 |
| Discount factor | 0.98 | 0.99 | 0.99 | 0.98 |
| Critic lr | $3 \times 10^{-4}$ | $3 \times 10^{-4}$ | $3 \times 10^{-4}$ | $3 \times 10^{-4}$ |
| Policy lr | $3 \times 10^{-4}$ | $3 \times 10^{-4}$ | $3 \times 10^{-4}$ | $3 \times 10^{-4}$ |
| temperature lr | $5 \times 10^{-3}$ | $5 \times 10^{-3}$ | $5 \times 10^{-3}$ | $5 \times 10^{-3}$ |
| HER Relabel | 80% | 80% | 80% | 80% |

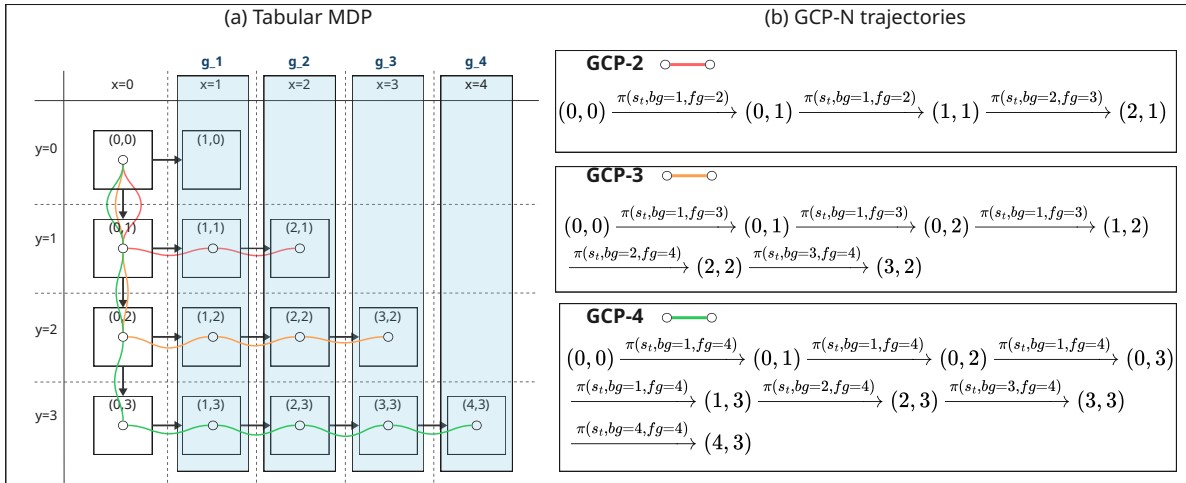

Figure 9: An example of tabular MDP where only conditioning the agent on the last goal can result in performing optimally. All agents conditioned on intermediate goals fail to discover the optimal sequence of actions.

### A.4 Failure Modes of GCP-N Agents

In this section, we present a simple tabular task designed to illustrate a failure mode that occurs for GCP-$N$ agents when $N$ is lower than the index of the final goal (here the maximum index is 4). The corresponding MDP is shown in Figure 9. States represent the agent position $(x, y)$ so the state space is $S \subseteq \mathbb{R}^2$. The agent starts at state $(0, 0)$ and can move deterministically along *directed* edges connecting cells. Additionally, a wait action allows the agent to remain in its current state. The goal space contains only the $x$-component of the state. Consequently, the planner outputs a sequence of intermediate goals corresponding to monotonically increasing $x$-values. For instance, if the agent starts from $(0, 0)$ and is tasked to reach $x = k$, the resulting plan is $(1, 2, \ldots, k)$. The reward function provides a value of $+1$ whenever an intermediate goal is reached, and 0 otherwise.

The trajectories executed by different GCP-$N$ agents when pursuing the final goal $x = 4$ are depicted in Figure 9. Looking at the trajectories for GCP-$N$ agents up to $N = 4$, we see that only the GCP-4 agent can successfully reach the final goal.

For instance, with GCP-2, the planner provides the goal sequence $(x = 1, x = 2, x = 3, x = 4)$. Initially, the policy is conditioned on $\pi(s = (0, 0), bg = 1, fg = 2)$. Since the agent was trained on transitions in which the final goal ($fg$) remains fixed throughout the episode, its optimal trajectory is the shortest path from $(0, 0)$ to reach $x = 1$, and then $x = 2$. This corresponds to the path: $[(0, 0) \rightarrow (0, 1) \rightarrow (1, 1) \rightarrow (2, 1)]$. Following this policy, GCP-2 moves from $(0, 0)$ to $(0, 1)$, and then to $(1, 1)$. Upon reaching $(1, 1)$, which corresponds to the first planner-provided subgoal ($x = 1$), the agent updates its conditioning to $\pi(s = (1, 1), bg = 2, fg = 3)$. At this point, however, the only reachable states are $(1, 1)$ and $(2, 1)$, and from these states there is no available path to reach $x = 3$. This prevents the agent from completing the planned trajectory. The agent thus transitions to $(2, 1)$, seeking the next intermediate reward. Once it reaches $(2, 1)$, the agent updates its conditioning to $(bg = 3, fg = 4)$. However, from the state $(2, 1)$, the only available action is the "wait" action, leaving the agent trapped in that state.

This specific example can easily be extended to any size to create failure cases for any GCP-$N$ agent where $N$ is smaller than the length of the planner's longest sequence of goals.

### A.5 Asymptotic performance of GCP-N Agents

In this section, we study whether a longer training budget allows GCP-$n$ variants with higher $N$ to achieve better performance.

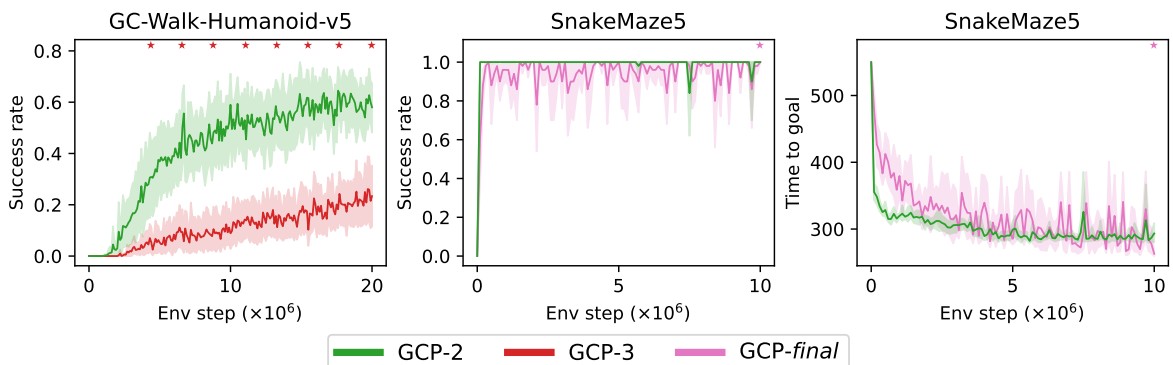

Figure 10: Success rate of GCP-2 and GCP-3 in *GC-Humanoid-Walk* with 20 millions steps, and of GCP-2 and GCP-*final* in *SnakeMaze5* with 10 millions steps. Each evaluation is performed 10 times, with results reported as the mean and 95% confidence interval (computed with bootstrapping) over 10 seeds. For *SnakeMaze5*, we also report time-to-goal, which better distinguishes agents (lower is better). Stars denote statistically significant differences (Welch T-test, p-value $< 0.05$) between GCP-2 and the algorithm of the corresponding color.

As shown in Figure 10, in *SnakeMaze5*, GCP-*final* performs similarly to GCP-2, though with higher variance, without significant statistical difference. To quantify this, we computed per-seed averages by taking the mean time to goal over the last 20 evaluation checkpoints for each of the 10 independent random seeds. This yielded 10 independent samples per method, each representing the final or asymptotic performance of a single seed. Comparing these samples using Welch's T-test, which accounts for unequal variances, resulted in a p-value of 0.73, indicating the absence of statistically significant difference between both methods.

In the case of *GC-Humanoid-Walk*, 20 million steps are still insufficient to reach asymptotic behavior, and due to computational constraints, we could not train any further. In principle, GCP-*final* could outperform GCP-2 in scenarios where anticipating further into the future is required. This is the case for instance in the *GC-Humanoid-Walk* task, where behaviors such as running efficiently toward a distant goal may only emerge if the agent plans sufficiently ahead. However, in practice, we did not observe such cases: the learned policies likely remain far from optimal, and GCP-2 achieves overall better performance.

## A.6   GCP-N success rate under higher UTD ratio

In this section, we evaluate whether increasing the update-to-data (UTD) ratio benefits GCP-*N* agents. As shown in Figure11, GCP-2 with UTD = 2 does not improve performance, and GCP-3 is negatively affected by the higher UTD ratio. Although higher UTD ratios can improve sample efficiency in some settings, this typically requires additional stabilizing modifications. For example, REDQ Chen et al. (2021) reports that large UTD ratios in SAC tend to amplify Q-value overestimation and instability; they counteract this using an ensemble of Q-functions, and in-target minimization across a random subset of Q functions from the ensemble.

In our work, we focus on the standard SAC algorithm as the low-level agent. Since high-UTD methods are not the focus of our study, we do not investigate such extensions.

## A.7   Evaluation of the value function during training

In this supplementary analysis, we qualitatively observe in Figure 12 the value function learned by GCP-*N* agents in *Dubins Hallway* using a single run. Our goal is to assess whether the learned critic $V_\theta^\pi$ matches a Monte Carlo estimate of the true value of the current policy $V^\pi$ along training, from a single start state $s_0$.

Interestingly, the Monte Carlo return converges to a higher value for policies conditioned on a more distant goal, as it includes more rewards for intermediate goals.

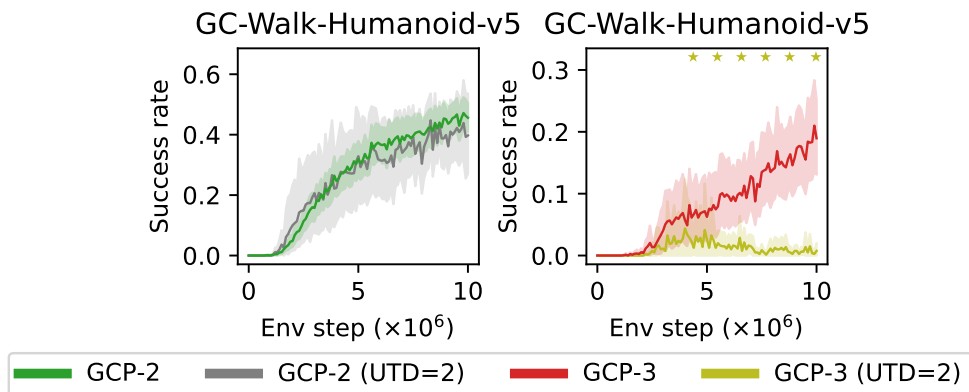

Figure 11: Performance comparison of GCP-2 and GCP-3 in *GC-Humanoid-Walk* for UTD values of 1 and 2. Each evaluation is performed 10 times, with results reported as the mean and 95% confidence interval (computed with bootstrapping) over 10 seeds. Stars denote statistically significant differences (Welch T-test, p-value $< 0.05$) between the compared algorithms.

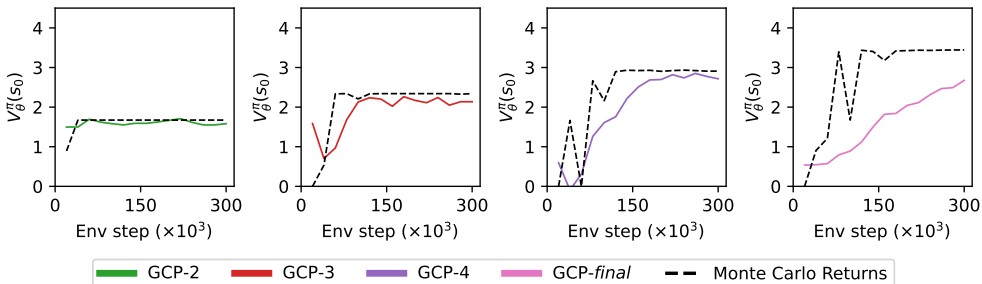

Figure 12: Evaluation of the value function learned by GCP-$N$ agents in *Dubins Hallway*. $V_\theta^\pi(s_0)$ denotes the critic's estimated value for the start state $s_0$ along training, for evaluation goals shown on the lower-left panel of Figure 2. At each evaluation step, we also report the mean of 10 Monte Carlo rollouts from $s_0$, which is a coarse estimate of the true value of the policy $V^\pi(s_0)$.

Regarding the critic value, the initial portion of the curves corresponds to a randomly initialized critic. As training progresses, smaller values of $N$ lead to a faster convergence of the critic toward an accurate estimate of the current performance of the policy. We see that GCP-$final$ is slower to converge, even though the agent reliably achieves the next five planned goals after 150K steps.

