# OpenReview forum: "A tale of two goals: leveraging short term goals performs best in multi-goal scenarios"
_TMLR — Accepted by TMLR_

### Review · Reviewer_ii1i · 2025-10-12

**Summary Of Contributions:**

This paper tackles the chaining issue in sequential low-dimensional goal-conditioned RL: an agent, achieving an intermediate goal, may arrive in a state that is incompatible with the next goal. The paper introduces an augmented MDP framework that integrates the current and final goal into the state and the transition function, enabling the agent to prepare for subsequent goals while reaching the current goal. The paper provides a solution method, GCP-N, that combines off-policy actor-critic (SAC) with modified Hindsight Experience Replay (HER). Through HER, the critic is trained to estimate action values conditioned on a wide range of goals. The actor is conditioned on the N-th goal, corresponding to selecting an action that optimizes reaching all intermediate goals up to the N-th one, with the extreme case being the final goal (GCP-final). The paper presents experimental results that demonstrate the impact of N, the better performance of GCP-2 against other choices of N and baselines across various environments, and an ablation on HER.

This paper is clearly written with substantive evidence for the proposed method in the experiments. The investigation into the choice of N was useful for justifying GCP-N, and the empirical methodology was overall sound. The framework is a tad complicated, as is typical for planning and abstraction, but clearly explained. Overall, the problem statement is clear, and the proposed solution addresses the problem. The augmented MDP framework that incorporates two goals into the state to preserve the Markov property is interesting, and GCP-N is a useful algorithmic contribution.

There are a few points to address.
1. SnakeMaze has 30 runs and the others have 10 runs. Using 10 runs for Gaussian CIs is a bit low. What was the reason for this difference, and could it be remedied?
2. There is a small change to SnakeMaze: “In de Lazcano et al. (2024), the agent’s velocity was capped between the range [-5, 5] m/s; we updated this limit to [-10, 10] m/s. ” What is the reason for this change?
3. Performance in Walk-Humanoid is poor for GCP-3 but not for GCP-2. The plots of value functions do help explain why this is the case (even though these plots are in Dubins Hallway). This might suggest that using more learning updates would be useful for GCP-3, GCP-4 and GCP-final. It might be worth testing this, seeing if values are more accurate and seeing if performance improves. If it does, then GCP-2 still has an advantage (computationally), but the advantage is more clearly a number of updates advantage rather than a learnability advantage.
4. You give the hyperparameters in the appendix, and they are sensible choices. But how were they chosen? I am also curious how important it was to have different gamma across these problems.

**Audience:**

Yes

**Audience Explanation:**

The practical problem being addressed and the proposed solution method in the paper may be of interest to those researching or applying sequential goal-conditional RL.

**Broader Impact Concerns:**

There are no concerns.

**Claims And Evidence:**

Yes

**Claims Explanation:**

The claims in the experiments section are generally well supported by the reported results.

**Requested Changes:**

Minor comments:
1. Pg 9, Fig 1: the font is small.
2. Page 10: There is one sentence that is a paragraph, starting “This is unexpected…”, which should be part of the previous paragraph. As a note, this is not that unexpected, and this sentence slightly contradicts the purpose of introducing the Nth version (which should presumably be easier to learn than GCP-final). Maybe what is surprising is that final performance is also worse, but really, that is hard to judge without running for longer or accounting for the fact that learning is harder and so the agent may need additional tricks.
3. In Section 3, it could be worth citing “Unifying task specification in reinforcement learning”, which uses transition-based gammas in an experiment in Taxi for an agent to learn an optimal policy between tasks, showing the importance of considering the next pick-up (goal) to get optimal policies.
4. In Section 5.1, the body text and the caption under Fig 1 are inconsistent in describing how the start state and the goal are sampled for each env. For example, in SnakeMaze5, “the agent starts in the green area” is written in the caption, but “the agent’s initial position and the goal are randomly sampled within the maze boundaries” is written in the body text.

---

> ### Author Response · Authors · 2026-01-01
>
> We thank the reviewer for their constructive review. We found all the points of the reviewer relevant and did our best to answer them with more complete experiments or additional studies. We believe it has helped us significantly improve our work. We uploaded a revised version of the manuscript where all the changes are highlighted in blue, and answered all the points individually below. We are ready to further interact with the reviewer if any question remains.
>
> In what follows we copy-paste the point of the reviewer in bold before providing our answer. Note that we added a new Figure 1, which results in shifting figure numbers in our answers below.
>
> **SnakeMaze has 30 runs and the others have 10 runs. Using 10 runs for Gaussian CIs is a bit low. What was the reason for this difference, and could it be remedied?**
>
> In the original setup, we used 30 runs in SnakeMaze to make the differences between methods more evident, especially in former Figure 2 (now Figure 3). We have now increased the number of runs to 30 in all environments, see the updated version of Figure 2->3.
>
> **There is a small change to SnakeMaze: “In de Lazcano et al. (2024), the agent’s velocity was capped between the range [-5, 5] m/s; we updated this limit to [-10, 10] m/s. ” What is the reason for this change?**
>
> In the original setup, capping the agent’s velocity at [-5, 5] m/s made it too easy to turn even at maximum speed, reducing the need to plan for future goals. Increasing the limit to [-10, 10] m/s requires the agent to accelerate and maneuver more strategically, making the task more challenging. Higher limits caused undesirable effects, like moving through walls. This explanation is added to the revised version of the manuscript p. 9.
>
> **Performance in Walk-Humanoid is poor for GCP-3 but not for GCP-2. The plots of value functions do help explain why this is the case (even though these plots are in Dubins Hallway). This might suggest that using more learning updates would be useful for GCP-3, GCP-4 and GCP-final. It might be worth testing this, seeing if values are more accurate and seeing if performance improves. If it does, then GCP-2 still has an advantage (computationally), but the advantage is more clearly a number of updates advantage rather than a learnability advantage.**
>
> We tested the hypothesis that additional learning updates might benefit GCP-N when $N$ is large, using two approaches.
>
> First, we extended the training budget. For SnakeMaze5, we increased the training budget from 300k to 10M steps (×30), and for GC-Humanoid-Walk, from 10M to 20M steps. In SnakeMaze5, GCP-final achieves similar performance to GCP-2, though with higher variance. In GC-Humanoid-Walk, 20M steps are still insufficient to reach asymptotic performance, and due to computational constraints, we could not extend training further. While it remains unclear whether GCP-final could eventually match GCP-2 with enough training in every case, the lower sample efficiency of GCP-N variants when $N$ is large becomes increasingly problematic as the environment complexity grows. We have added a new section in the appendix (Section A.5, page 19) to further discuss this point.
>
> Second, we attempted to increase the update-to-data (UTD) ratio during training. However, with our current vanilla SAC implementation, simply raising the UTD ratio does not improve performance; achieving stable learning would require additional algorithmic modifications. A short description of this result is included in Appendix A.6, page 22.

---

> ### Author Response · Authors · 2026-01-01
>
> **You give the hyperparameters in the appendix, and they are sensible choices. But how were they chosen? I am also curious how important it was to have different gamma across these problems.**
>
> The hyper-parameters used for the SAC backbone in all methods were mainly taken from the original SAC paper [1]. We only increased the neural network size for more complex tasks, as in [2]. We added a sentence in Appendix A.3 to account for this fact.
>
> **Choice of discount factors**:
> We did not perform an extensive sensitivity study on the discount factor. Instead, we followed values used in previous work. Specifically, we used 0.99 as in [1], and reused task-specific values when available. In GC-Humanoid-Walk we used 0.98 as in [2]. In Dubins-Hallway, we initially used 0.9 as in [2], but increased it to 0.98 to improve the readability of value plots in the former Figure 3 (now Figure 4). Performances were similar in both cases. For SAC+HER, that directly targets the final goal, we experimented with a higher discount factor (0.995) in long-horizon tasks. This improved performance in GC-Cartpole and SnakeMaze5, while having no effect on GC-Humanoid-Walk. Preliminary studies with either SAC+HER+SEQ or GCP-N showed no benefits from increasing the discount factor.
>
> **Requested Changes**:
> About the minor comments of the reviewer, we did the following:
>
> 1) We increased the font size as requested.
>
> 2) We removed the sentence starting with “This is unexpected…” as we agree with the reviewer.
>
> 3) [3] introduces a transition-based discount $\gamma(s,a,s’)$, allowing termination and discounting to depend on specific transitions and thereby decomposing a single MDP into subtasks. With a soft discount formulation, this framework permits partial propagation of value across subtask boundaries, enabling the agent to account for rewards that occur beyond the current subtask. We added a short section 3.5 (pp. 6-7) where we discuss the relationship between this work and ours, together with the relationship to [2] (see above).
>
> 4) The question about inconsistencies between the captions and text helped us realize that the presentation of training and evaluation conditions were insufficiently clear. We added a clearer description of these aspects for all environments.
>
> [1]: Haarnoja, T., Zhou, A., Hartikainen, K., Tucker, G., Ha, S., Tan, J., ... & Levine, S. (2018). Soft actor-critic algorithms and applications. arXiv preprint arXiv:1812.05905.
>
> [2]: Chenu, A., Perrin-Gilbert, N., & Sigaud, O. (2022, October). Divide & conquer imitation learning. In 2022 IEEE/RSJ International Conference on Intelligent Robots and Systems (IROS) (pp. 8630-8637). IEEE.
>
> [3]: White, M. (2017, July). Unifying task specification in reinforcement learning. In International Conference on Machine Learning (pp. 3742-3750). PMLR.

---

### Review · Reviewer_hR8T · 2025-11-04

**Summary Of Contributions:**

This paper addresses the problem of multi-goal conditioned reinforcement learning, which extends the goal-conditioned setting to the case where an agent must reach a sequence of goal locations in an MDP. The main contributions of the paper are:

1) The observation that agents conditioned on a single goal may fail to ensure that each intermediate goal is reached in a way that remains compatible with reaching the rest of the sequence of goals;

2) A generalization of goal-conditioned policies (GCP) in which a policy is conditioned not only on the next current goal, but also on the N-th goal from a provided sequence of goals given by a fixed planner; and

3) A practical implementation of this method in an actor-critic framework that extends Hindsight Experience Replay (HER) to the multi-goal setting. The experimental results suggest that GCP-2, i.e., conditioning only the two most immediate goals, provides the best performance in the benchmarks.

Strengths:
- The idea is simple and easy to incorporate into existing goal-conditioned methods.

Weaknesses:
- Although the authors acknowledge the limitation of using a fixed oracle planner, this point is of high relevance and deserves a more in-depth discussion. Assuming a fixed planner is a significant assumption and may limit the method's applicability to the community.
- The paper relies solely on empirical results, while some of the claims could also be backed up by theoretical/mathematical results.

**Audience:**

Yes

**Audience Explanation:**

I believe this paper would gather interest due to the fact that goal-conditioned RL is a highly studied topic. Moreover, the proposed technique is straightforward to implement, and extending HER to multi-goal settings is also an interesting contribution.

**Broader Impact Concerns:**

NA.

**Claims And Evidence:**

Yes

**Claims Explanation:**

The main claims of the paper are backed up by empirical results. As discussed below, the paper could be strengthened by a more mathematical/theoretical analysis that can better characterize the problem and the trade-off resulting from the method’s hyperparameter $N$.

**Requested Changes:**

- A smaller value of $N$ may lead to actions that are suboptimal with respect to the entire sequence of goals. However, the shorter horizon reduces sensitivity to compounding errors over time. It would significantly strengthen the paper if the authors could provide a more mathematical/theoretical justification of this claim, e.g., a novel theorem characterizing the trade-off of choosing a good value for $N$.

- I suggest expanding the discussion on the limitations of using a fixed planner. What are the challenges associated with learning a planner simultaneously? Do you think this problem is tractable? If not, in which types of scenarios is it reasonable to assume access to an oracle goal planner?

- Should the set $G$ be part of the definition of $M_{cg}$ in Section 2.2.?

- In Fig. 2, why not include a standard GC policy (GCP-1) that is conditioned only on the current goal? Is it possible that it would also work well, showing that the benchmarks do not have the property being evaluated, which is “balance immediate progress toward bg with preserving the feasibility of future goals”?

- In Section 5.2, it is not surprising that smaller values of $N$ lead to higher simple efficiency. However, it would be relevant to investigate if higher values of $N$ could lead to better asymptotic performance if trained for a sufficiently long time. For instance, in SnakeMaze5, could GCP-4 outperform GCP-2 in the long term? Would it be possible to construct a scenario in which it is strictly necessary to condition on $N>2$ goals?

---

> ### Author Response · Authors · 2026-01-01
>
> We thank the reviewer for their constructive review. We found all the points of the reviewer relevant and did our best to answer them. We believe it has helped us significantly improve our work by shedding more light on the value of the different variants we have studied. We uploaded a revised version of the manuscript where all the changes are highlighted in blue, and answered all the points individually below. We are ready to further interact with the reviewer if any question remains.
>
> In what follows we copy-paste the point of the reviewer in bold before providing our answer. Note that we added a new Figure 1, which results in shifting figure numbers in our answers below.
>
> **A smaller value of N may lead to actions that are suboptimal with respect to the entire sequence of goals. However, the shorter horizon reduces sensitivity to compounding errors over time. It would significantly strengthen the paper if the authors could provide a more mathematical/theoretical justification of this claim, e.g., a novel theorem characterizing the trade-off of choosing a good value for N.**
>
> To answer the reviewer’s point, we need to distinguish two claims:
> - Claim 1: within our framework, there can be hard environments in which conditioning the policy on intermediate goals rather than on the final one can prevent convergence to the optimal behavior.
> - Claim 2: when the environment is simpler, conditioning on more immediate goals generally performs better than conditioning on later goals.
>
> We agree with the reviewer that providing a more mathematical/theoretical justification of these claims would strengthen the paper. To back up Claim 1, we now present in Appendix A.4  a tabular MDP example where conditioning the policy on intermediate goals is unquestionably suboptimal. We believe that exhibiting such a counterexample is enough to back up Claim 1. We are aware that providing a more general proof might be more valuable, but would probably require limiting assumptions. Regarding Claim 2, similarly, we believe that providing a rigorous justification would require switching to simplified tabular settings and restricting the class of function approximators we use, which would drive us too far away to our context based on deep RL algorithms whose mathematical foundations are limited. While theoretical proofs can offer valuable insights, our interest lies primarily in empirical studies under realistic conditions, where our methods can demonstrate their practical relevance.

---

> ### Author Response · Authors · 2026-01-01
>
> **I suggest expanding the discussion on the limitations of using a fixed planner. What are the challenges associated with learning a planner simultaneously? Do you think this problem is tractable? If not, in which types of scenarios is it reasonable to assume access to an oracle goal planner?**
>
> We answer the reviewer’s questions in several subparts: first, the challenges of learning both the high and low levels simultaneously, then more explicit motivations for using a fixed planner.
>
> **Challenges of learning both the high and low levels simultaneously**
>
> Many works in the HRL literature show that simultaneously learning a high-level policy (a.k.a a planner) and a low-level policy can also result in instabilities in the high-level MDP (see e.g. HAC [1], HIRO [2]). Besides, there are many options to learn these high-level and low-level policies, as well as many dedicated mechanisms to prevent the resulting instabilities. Rather than trying to address the full HRL framework and facing this huge combinatorial of options, we deliberately chose to address the more local question of learning the low-level policy given some high-level planner, where we can reach more solid conclusions.
>
> To be more specific about the instabilities that may arise with GCP-N when learning both the high and low levels simultaneously, if the planner evolves over time, the low-level MDP can become non-stationary, depending on how the behavioral goal dynamics are defined. When the behavioral goal is fixed throughout the episode (as in SAC+SEQ+HER), the dynamics remain stationary. In contrast, if the behavioral goal switches to the next goal provided by the planner (as in GCP-N), each change in the planner modifies the low-level MDP and can introduce learning instabilities. However, the less the agent prepares for future goals, the fewer planner-dependent behavioral goal transitions. An extreme case occurs when the agent is conditioned on the next two goals of a plan: in this case, the evolution of behavioral goals becomes planner-independent. In such cases, a GCP-N agent conditioned on $\pi(..., bg=A, fg=B)$ can only switch to $\pi(..., bg=B, fg=B)$ once goal $A$ has been reached. Consequently, GCP-N with smaller $N$ are likely to be more stable. We clarified these points in the conclusion of the paper.
>
> **Fixed planner**
> As a consequence of isolating the low-level policy learning problem, we consider a fixed planner. In our work, we are using an oracle planner, which has limitations as it requires expert knowledge. While it may not always be feasible, it can be a reasonable assumption in certain domains. For instance, in our experiments, the oracle planner simply accounts for obstacles (walls) and generates $(x,y)$ intermediate goals via a shortest-path algorithm, **without modeling the full agent dynamics**. The availability of such an oracle planner is a reasonable assumption in navigation tasks with known environment geometry. In the more general case where a high-level policy generates the plan, inefficient goal generation may raise additional issues. We leave the study of such situations for future work, and we added a sentence in the conclusion of the revised version to stress this limitation of our work.
>
> [1] Andrew Levy, George Konidaris, Robert Platt, and Kate Saenko. Learning multi-level hierarchies with hindsight. In Proceedings of International Conference on Learning Representations, 2019.
>
> [2] Ofir Nachum, Shixiang Gu, Honglak Lee, and Sergey Levine. Data-Efficient Hierarchical Reinforcement Learning, October 2018.

---

> ### Author Response · Authors · 2026-01-01
>
> **Should the set G be part of the definition of M_{gc} in Section 2.2.?**
>
> We thank the reviewer for this suggestion. In $M_{gc}$, we define $S_{gc} = S \times G$, and we have updated the manuscript to make the inclusion of $G$ more explicit.
>
> **In Fig. 2, why not include a standard GC policy (GCP-1) that is conditioned only on the current goal? Is it possible that it would also work well, showing that the benchmarks do not have the property being evaluated, which is “balance immediate progress toward bg with preserving the feasibility of future goals”?**
>
> We did not explicitly include a GCP-1 variant in our experiments. In such a case, the policy would be conditioned on two identical goals, $\pi(s,bg_t,bg_t)$, which effectively means that the agent optimizes only for the current goal. We expect this variant to behave very similarly to the SAC+HER+SEQ baseline, where the policy is conditioned solely on the current goal, $\pi(s,bg_t)$. In both cases, the actor focuses exclusively on achieving the current goal without preparing for future ones. As shown in Section 5.3, SAC+HER+SEQ fails to achieve satisfactory performance in the Dubins Hallway environment: the reported 20% success rate corresponds to reaching only the first of the five evaluation goals depicted in former Figure 1 (now Figure 2). To clarify this point, we have added p. 11 a footnote explaining the absence of GCP-1, and directing the reader to Sections 5.3 and 5.4 where SAC+HER+SEQ results are discussed.
>
> **In Section 5.2, it is not surprising that smaller values of N lead to higher simple efficiency. However, it would be relevant to investigate if higher values of N could lead to better asymptotic performance if trained for a sufficiently long time. For instance, in SnakeMaze5, could GCP-4 outperform GCP-2 in the long term? Would it be possible to construct a scenario in which it is strictly necessary to condition on N>2 goals?**
>
> We have added a new Appendix (A.5 / p. 21) investigating whether higher values of $N$ could lead to better asymptotic performance. For the SnakeMaze5 task, we increased the training budget from 300K to 10M steps (×30), and for GC-Humanoid-Walk, from 10M to 20M steps. In SnakeMaze5, GCP-final achieves similar performance to GCP-2, though with higher variance. In GC-Humanoid-Walk, 20M steps are still insufficient to reach asymptotic performance, and due to computational constraints, we could not train further. We believe this is enough to claim that GCP-2 is superior to GCP-final if one considers reasonable budget constraints.
>
> In principle, GCP-final could outperform GCP-2 in scenarios where anticipating further into the future is necessary. For example, in the GC-Humanoid-Walk task, behaviors such as running efficiently toward a distant goal may only emerge if the agent plans sufficiently ahead.
>
> From a theoretical standpoint, we also added section (A.4, page 20) introducing a simple tabular task that illustrates a failure mode that occurs for GCP-$N$ agents when $N$ is lower than the index of the final goal.

---

> > ### Comment · Reviewer_hR8T · 2026-01-22
> >
> > I thank the authors for carefully answering my questions and concernes and for taking the reviewers' suggestions into account in the updated manuscript.
> >
> > I do not have any remaining concerns/questions.

---

### Review · Reviewer_teVv · 2025-12-25

**Summary Of Contributions:**

The primary contribution of this paper is a method for a goal-conditioned policy that, instead of considering a single subgoal, takes two subgoals as input: the next subgoal and a subgoal N steps in the future. The paper also presents a modification to hindsight experience replay to handle these two subgoal policies.

The paper's strength is that it offers a relatively simple approach to overcoming the limitations of a subgoal space that considers only position, thereby making it difficult for the agent to transition from one goal to the next.

The paper's weaknesses lie in its clarity and the lack of support for its claims. In terms of clarity, the problem area, assuming that the abstract planner cannot use a subgoal space that would enable a myopic goal-conditioned policy to perform well, is not well motivated; it is, however, an acceptable problem area. There are also numerous imprecise statements in the paper; some are listed in the claims section.  The writing also lacks conciseness and a clear flow of ideas that build on one another; it is at times repetitive. These writing issues can be fixed.

**Additional Comments:**

- It should be stated how $k_1$ and $k_2 are chosen in the method.

**Audience:**

No

**Audience Explanation:**

I say it is not of interest due to the lack of support for its claims. Future versions of this work, with problems fixed, could be of interst to the community.

**Broader Impact Concerns:**

No broader impact concerns

**Claims And Evidence:**

No

**Claims Explanation:**

Below is a list of claims I found in the paper and a statement regarding the sufficiency or lack of evidence. Overall, the claims made are not supported by direct evidence or are not scoped to the evidence provided.

**Performance-related claims**:

For all performance-related claims, there are several key problems: 1) there is a lack of a clear metric used to make the claim and perform statistical analysis, judgments of superior or faster optimization are based on subjective observations of learning curves, 2) it is unclear what methods are used to compute confidence intervals and the number of trials is likely too low to be sufficient for rely on them, 3) there is no correction for multiple comparisons, 4) lastly all algorithms are run with the same hyperparameters, this means one algorithm will be better than another just do to the choice of hyperparameters. No discussion is given on how the hyperparameters are chosen or tuned and this process was not taken into account for the uncertainty of performance. The claims are not scoped to account for this narrow interpretation of performance.

1. “in most cases, preparing to reach the next two goals improves stability and sample efficiency over all other approaches.”

  Source: abstract

  Issues: stability is undefined and undiscussed in the paper. Sample efficiency is also undefined.

2. “Our results suggest that conditioning the policy on the two most immediate successive goals (N = 2) is
the most efficient variant. This approach outperforms both myopic agents and non-sequential agents.”

  Source: end of introduction

  Issues: same as given above.

3. “As Figure 2 shows, smaller values of N tend to yield both higher sample efficiency and better performance
in all environments tested. In particular, the best performance is observed for N = 2,”

  Source: Section 5.2, paragraph 2.

  Issue: same as above.


4. “Our method, GCP-2, consistently achieves the highest or comparable success rates in all environments. It
is the only method that achieves nontrivial success in the GC-Humanoid-Walk task, and it also achieves the
best sample efficiency in Dubins Hallway.”

Source Section 5.3, last paragraph

Issue: Same as above. What does nontrivial success mean? This is not a defined or precise term.



**Claims in regard to conditioning on a future subgoal lead to optimizing actions for intermediate goals**:

The paper makes numerous claims suggesting that having a single future subgoal will lead to the selection of actions that are optimal for the intermediate subgoals. This claim does not make sense and is not supported by the evidence in the paper. Consider an agent in a 1D environment where the final goal is to go to the right, but the first half of the subgoals are all to the left. Conditioning on any goal that is to the right tells the agent nothing about how it should go to the left or when it should prepare to go to the right. If the agent were instead given a sequence of N subgoals, then it could make decisions to achieve all of them in sequence, but this is not what is proposed or what is discussed.

The claims listed below are taken from the paper and are stated without direct or sufficient evidence.

5. “Thus, the policy must balance immediate progress toward bg with preserving the feasibility of future goals.”

  Source: Section 4.3, paragraph 2.

6. “Given that the GCP-N policy is optimized on xGC-MDP, it selects an action that optimizes reaching all intermediate goals up to the N-th one…N = final has the lowest performance and sample efficiency.“

  Source: Section 4.3, second to last paragraph

7. “only when using GCP-final the agent can prepare for the entire sequence of future goals.”

Source: Section 5.2, third paragraph

8. “In such cases, the actor primarily learns to select actions that prepare the agent for reaching the first few intermediate goals, while the influence of more distant goals becomes negligible.“

Source: Section 5.2, last paragraph


**Other claims**:

The following claims are made in the paper but are not covered by the two categories above.

9. “However, the shorter horizon reduces sensitivity to compounding errors overtime”

Source: Section 4.3, second to last paragraph

Issue: What are the comounding errors? This is the only time this was discussed in the paper, and it is unclear what it is referencing or what evidence is used to make the claim

10. “This indicates that the critic trained with sac has only propagated the value from the first few goals of the sequence instead of all goals up to the final one.”

Source: Section 5.2, second to last paragraph

Issue: This plot shows the difference in prediction error and does not directly show how value is propagated across subgoals. Thus, there are multiple ways this difference between the different values of N could have been observed. For example, the other methods (N=2,3) were able to reach the final goal faster, so they have had more time to learn the value function for a policy that goes through all the goals. This faster learning could just be due to beneficial hyperparameters or some factor other than value propagation, this this claim is unsupported. Looking at value estimates and propagation during learning would be interesting and a possible route to explain the behavior. However, this experiment alone is insufficient to draw such a conclusion.

There is another issue, where it is claimed that "In deterministic environments, the Monte Carlo return and Vπ coincide," but this is only true if the policy is deterministic. In this case, the policy is not deterministic, and the value function cannot be evaluated from a single episode. Furthermore, if a deterministic policy was created from a stochastic policy, then the value function would have been trained on the data from the stochastic policy, and there is no reason to assume it should match unless the policy is deterministic.

11. “The agent does not differentiate between intermediate and final goals, leading it to decelerate before each goal to ensure it can stabilize there.”

Source: Section 5.4, paragraph 2.

Issue: This is likely true, but there is no evidence for this claim. It would make sense that this is the desired behavior, but this behavior needs to be measured.

12. “In SnakeMaze5, sac+her+seq instead reaches goals at high speed. However, this behavior makes it difficult to turn, resulting in bouncing off walls (Figure 6 (a)). As shown in Figure 6(b), the time the agent needs to reach the final goal is longer than with GCP-2.”

Source: Section 5.4

Issue: Similar to the other statement, there is no direct evidence that it reaches the goals at higher speeds. The paths shown in Figure 6 (a) do not show velocity. The paths shown in Figure 6 (a) would be evidence that conditioning on the second goal helps the policy over just having the first goal, but not sufficient on its own. This is because, even without any goal information, a good policy should be able to navigate to the chosen goal as quickly as possible. This suggests (i.e., needs investigation) that two or more subgoals used to select actions may influence the policy (change the objective surface) to prefer a smooth path between goals, rather than simply seeking to reach a single goal as quickly as possible. This could be a route to understanding Figure 6 (a), but it is not the only one.

*Question: Were these paths smoothed in the plotting?* They look unusually smooth for an RL agent.

**Requested Changes:**

**Critical changes**:

1. All claims are either properly supported or scoped to the available evidence. For many of these claims, it would likely require new experiments that look beyond simple performance metrics. If ALL claims currently in the paper were only scoped to the available evidence, the paper would no longer have interesting results.


2. It is unclear why one would want to condition on a subgoal N>2 steps in the future and not include the subgoals between the current one and the Nth one. The value of N=2 is the only choice that actually chains together subgoals. It should be made clear why N>2 is even considered or could be beneficial. The stated reasons (as discussed above) do not follow.

**Important but not critical**:

3. It should be made clear why we are looking at planners that provide bad subgoals. For example, in the walker problem, the sequence of subgoals is just a series of vectors to move straight ahead. Why not have just a single goal for this problem to reach the final position? There are many examples (and I have personal experience) that make the humanoid move to different positions and require the agent to change directions. What does providing a single

---

> ### Author Response · Authors · 2026-01-01
>
> We thank the reviewer for their request for more clarity and more rigor in the expression of our claims. We did our best to satisfy these requests to improve our paper. We uploaded a revised version of the manuscript where all the changes are highlighted in blue, and answered all the points individually below. We are ready to further interact with the reviewer if any question remains.
>
> In what follows we copy-paste the point of the reviewer in bold before providing our answer. Note that, to answer some of the points of the reviewer, we added a new Figure 1, which results in shifting figure numbers in our answers below.
>
> **For all performance-related claims, there are several key problems: 1) there is a lack of a clear metric used to make the claim and perform statistical analysis, judgments of superior or faster optimization are based on subjective observations of learning curves, 2) it is unclear what methods are used to compute confidence intervals and the number of trials is likely too low to be sufficient for rely on them, 3) there is no correction for multiple comparisons, 4) lastly all algorithms are run with the same hyperparameters, this means one algorithm will be better than another just do to the choice of hyperparameters. No discussion is given on how the hyperparameters are chosen or tuned and this process was not taken into account for the uncertainty of performance. The claims are not scoped to account for this narrow interpretation of performance.**
>
> Regarding metrics, we agree with the reviewer that  “performance” could sometimes denote the episode return, the success rate, or the time to goal.  We have clarified all uses of the term throughout the paper. See in particular the penultimate paragraph of the Introduction : “(4) Given the possibility of preparing up to $N$ goals in advance, we compare how different choices of $N$ affect different performance criteria: the episode return, the success rate or the time to goal, depending on the environment.” See also the new Figure 3, for instance.
>
> We agree with the reviewer that claims based on subjective observations of learning curves were unsatisfactory, so we added standard statistical tests (Welch T-test, see [1]) to back-up our claims in former Figs 2, 4, 6, 7, 9 and 10 (now Figs 3, 5, 7, 8, 10 and 11). As we are only performing one-to-one algorithm comparisons, we do not incorporate Bonferroni correction, but we are ready to change this if the reviewer feels it is necessary. The confidence intervals are constructed by taking a percentile interval of the bootstrap distribution and we increased the number of seeds to 30 in all comparisons to baselines.
>
> About hyper-parameters, our approach is the following. The hyper-parameters used for the SAC backbone in all methods were mainly taken from the original SAC paper [2]. We only increased the neural network size for more complex tasks, as in [3]. We added a sentence in Appendix A.3 to account for this fact.
>
> Similarly, about discount factors, we did not perform an extensive sensitivity study on the discount factor. Instead, we followed values used in previous work. Specifically, we used 0.99 as in [2], and reused task-specific values when available. In GC-Humanoid-Walk we used 0.98 as in [2]. In Dubins-Hallway, we initially used 0.9 as in [2], but increased it to 0.98 to improve the readability of value plots in the former Figure 3 (now Figure 4). Performances were similar in both cases. For SAC+HER, that directly targets the final goal, we experimented with a higher discount factor (0.995) in long-horizon tasks. This improved performance in GC-Cartpole and SnakeMaze5, while having no effect on GC-Humanoid-Walk. Preliminary studies with either SAC+HER+SEQ or GCP-N showed no benefits from increasing the discount factor.
>
> [1]: Colas, C., Sigaud, O., & Oudeyer, P. Y. (2019). A hitchhiker's guide to statistical comparisons of reinforcement learning algorithms. arXiv preprint arXiv:1904.06979.
>
> [2]: Haarnoja, T., Zhou, A., Hartikainen, K., Tucker, G., Ha, S., Tan, J., ... & Levine, S. (2018). Soft actor-critic algorithms and applications. arXiv preprint arXiv:1812.05905.
>
> [3]: Chenu, A., Perrin-Gilbert, N., & Sigaud, O. (2022, October). Divide & conquer imitation learning. In 2022 IEEE/RSJ International Conference on Intelligent Robots and Systems (IROS) (pp. 8630-8637). IEEE.

---

> ### Author Response · Authors · 2026-01-01
>
> **1) “in most cases, preparing to reach the next two goals improves stability and sample efficiency over all other approaches.”**
> **Issues: stability is undefined and undiscussed in the paper. Sample efficiency is also undefined.**
>
> The final sentence of the abstract was: “By evaluating policies trained with an off-policy actor-critic algorithm on both the standard goal-conditioned MDP framework and ours, we show that, in most cases, preparing to reach the next two goals improves stability and sample efficiency over all other approaches.”
>
> We agree that we did not define a stability metric and that, although sample efficiency is a well-understood notion in the RL literature that does not need to be defined any more (see e.g. [4] and many others), our use of it was quite vague in the above statement. In the revised version, we replaced the above sentence with:
>
> “2) By evaluating policies trained with an off-policy actor-critic algorithm on both the standard goal-conditioned MDP framework and ours in these tasks, we show that learning policies conditioned on the next two goals generally require less interaction data than all the other types to reach the same level of performance or better.”
>
> [4]: Ye, W., Liu, S., Kurutach, T., Abbeel, P., & Gao, Y. (2021). Mastering atari games with limited data. Advances in neural information processing systems, 34, 25476-25488.
>
> **3) “Our results suggest that conditioning the policy on the two most immediate successive goals (N = 2) is the most efficient variant. This approach outperforms both myopic agents and non-sequential agents.”**
>
> **Issues: same as given above.**
>
> Again, we replaced the above sentence with: “Our results suggest that policies conditioned on the two most immediate successive goals ($N = 2$) can generally be learned from less interaction data than myopic agents and non-sequential agents to reach the same level of performance or better.”
>
> **“As Figure 2 shows, smaller values of N tend to yield both higher sample efficiency and better performance in all environments tested. In particular, the best performance is observed for N = 2,”**
>
> **Issue: same as above.**
>
> We replaced the above sentence with: “As Figure 3 shows, learning a policy with smaller values of N tends to perform equally or better in fewer interactions with the environment. In particular, the fastest improvement is observed for $N = 2$, corresponding to the setting in which the agent is conditioned on the next two upcoming goals, while it tends to be the slowest for $N = final$.”
>
> **4) “Our method, GCP-2, consistently achieves the highest or comparable success rates in all environments. It is the only method that achieves nontrivial success in the GC-Humanoid-Walk task, and it also achieves the best sample efficiency in Dubins Hallway.”**
>
> **Issue: Same as above. What does nontrivial success mean? This is not a defined or precise term.**
>
> We replaced the above sentence with: “Our method, GCP-2, consistently achieves the highest or comparable success rates in all environments. It is the only method whose success rate starts to climb over 1% after 2M steps in the GC-Humanoid-Walk task (getting above 40% after 10M steps), and gets close to 100% in Dubins Hallway in less than 150K steps.”

---

> ### Author Response · Authors · 2026-01-01
>
> **Claims in regard to conditioning on a future subgoal lead to optimizing actions for intermediate goals**
>
> We believe the reviewer partly misunderstood our point. This helped us realize that introducing a didactic figure would significantly clarify the issue we are addressing in our work. We added Figure 1 which we hope makes the story clearer and revised the Methods section accordingly. Below we answer in more detail the reviewer’s points, but we think the reviewer should take a look at Figure 1 first. We hope the reviewer will find our point clearer now and we are ready to further discuss this otherwise.
>
> **The paper makes numerous claims suggesting that having a single future subgoal will lead to the selection of actions that are optimal for the intermediate subgoals. This claim does not make sense and is not supported by the evidence in the paper.**
>
> For clarification, our claims concern an agent conditioned on both the current and a future subgoal and optimizing the xGC-MDP. In this context, we claim that an optimal policy would select actions that also support reaching intermediate subgoals. This statement is about the ideal behavior of such an optimal policy, rather than empirical results. Although GCP-N agents are trained with SAC and may not act optimally, the optimal policy serves as a useful reference for interpreting their behavior. Further justification for these claims is discussed in our next response.
>
> **Consider an agent in a 1D environment where the final goal is to go to the right, but the first half of the subgoals are all to the left. Conditioning on any goal that is to the right tells the agent nothing about how it should go to the left or when it should prepare to go to the right.**
>
> We thank the reviewer for this comment. In the xGC-MDP framework, the agent is conditioned on both the current behavioral goal and some next goal, which can be the final goal. We agree that this conditioning alone is not sufficient for the agent to determine how to optimize intermediate goals.
>
> The key distinction lies in the optimization objective of the xGC-MDP. For clarity, we focus here on the GCP-final formulation (see Figure 1(3)), which directly optimizes the xGC-MDP objective (the GCP-N case follows as a special case). For simplicity, we also assume here that reaching a goal is terminal (a general formulation is provided in the paper).
>
> In a standard GC-MDP, the optimization objective assumes a fixed goal. The policy is either conditioned on the final goal (in which case intermediate goals are not used, see Figure 1(1)), or the current behavioral goal (in which case the episode terminates upon reaching the first goal, ignoring the subsequent evolution of goals).
>
> In contrast, the xGC-MDP explicitly models the dynamical evolution of the behavioral goal and assigns a reward of +1 for each intermediate goal if reached in the prescribed order, terminating only when the final goal is achieved. Under this objective, a policy that reaches $N$ future goals has value close to $N$ (ignoring discounting), whereas policies that skip or fail to reach intermediate goals obtain a strictly lower return. Consequently, solving the xGC-MDP favors policies that sequentially achieve all goals in the plan, including those that may initially require moving away from the final goal, as in the reviewer’s example.
>
> We revised Section 4.2 to clarify this formalization and added Figure 1 illustrating the different settings, and particularly how goals evolve over time and how the goal horizon differs across the considered variants.
>
> **If the agent were instead given a sequence of N subgoals, then it could make decisions to achieve all of them in sequence, but this is not what is proposed or what is discussed.**
>
> We agree with the reviewer that conditioning the agent on the full sequence of remaining subgoals would also define a valid MDP, and we briefly discuss this variant in Section 4.2. In this formulation, the agent is conditioned on the sequence of remaining goals $\boldsymbol{g}_t$ and receives a reward of +1 each time the first goal in the sequence is reached, after which this goal is removed. However, the main difficulty is that the number of goals along the path can vary from one path to another.
> We conducted preliminary experiments with this approach, using RNN and transformer-based architectures to handle variable-length goal sequences. However, we observed strong sensitivity to architectural choices and unstable performance, failing to solve SnakeMaze5. We hypothesize that conditioning on long sequences of goals increases the difficulty of generalization, as the actor and critic must handle a much larger and more variable input space, which in turn can lead to out-of-distribution behavior at inference time. Pursuing this direction would require extensive architecture-specific tuning as well as careful design of relabeling strategies; we therefore consider these experiments beyond the scope of the present work.

---

> > ### Comment · Reviewer_teVv · 2026-02-02
> >
> > Ah ok, I believe I better understand the argument now (added text in Section 4.2 helped). However, I think the text is unclear. My understanding is that GCP-N optimizes for following a sequence of N behavior goals and uses a policy conditioned on the next goal and the N-th goal. The policy is then used in an environment with K>N behavior goals, and once a behavior goal is achieved, the problem becomes a K-1 behavior goal problem. So the GCP-N method should work well when 1) the agent can follow a sequence of N behavior goals,  2) the agent only needs to be myopic with respect to N behavior goals, and 3) the policy does not cause the agent to be in an unfavorable state when achieving a behavior goal and the next fg switches to the next behavior goal. With N=2 the policy is only optimized for following a sequence of two behavior goals and thus should provide a smooth transition between behavior goals since actions are always selected to enable the agent get to the next behavior goal. However, when the last goal is more than N behavior goals away, the policy may take actions that prevent the agent from reaching it (violating assumption 2).
> >
> > Assuming my understanding is correct, I think Sections 2 and 4 should be rewritten to make it clearer and more concise.

---

> ### Author Response · Authors · 2026-01-01
>
> **The claims listed below are taken from the paper and are stated without direct or sufficient evidence.**
>
> **5) “Thus, the policy must balance immediate progress toward bg with preserving the feasibility of future goals.”**
>
> The sentence before the one pointed out by the reviewer was “In order to maximize the objective function, the agent should reach $bg$ in a configuration from which it can achieve the rest of the sequence.” In fact, we believe this previous sentence already captures what we meant, and we removed the additional one that the reviewer has pointed out, as we agree this second sentence was making a more vague point.
>
> **6) “Given that the GCP-N policy is optimized on xGC-MDP, it selects an action that optimizes reaching all intermediate goals up to the N-th one…N = final has the lowest performance and sample efficiency.“**
>
> The excerpt from the reviewer mixes a sentence from Section 4.3 and another from Section 5.2.
>
> The complete paragraph in Section 4.3 in the submitted version was “Given that the GCP-N policy is optimized on xGC-MDP, it selects an action that optimizes reaching all intermediate goals up to the $N$-th one. When $N$ is large, the actor must perform actions that are compatible with the achievement of many future goals, enhancing the probability that the action is optimal when considering the whole sequence. In contrast, a smaller value of $N$ limits the optimization horizon and may lead to actions that are suboptimal with respect to the entire sequence. However, the shorter horizon reduces sensitivity to compounding errors over time, and the agent is more likely to have encountered training transitions that involve nearby goals rather than distant ones”.
>
> We replaced this paragraph with: Given that the GCP-N policy is learned on xGC-MDP, it selects an action that optimizes reaching all intermediate goals up to the $N$-th one. When $N$ is large, the actor must perform actions that are compatible with the achievement of many future goals, increasing the probability that the action is optimal when considering the whole sequence. *In contrast, a smaller value of $N$ limits the optimization horizon and may lead to actions that may not reach the final goal in the most difficult cases. However, the shorter horizon improves the estimation of the value function [4]*, and the agent is more likely to have encountered training transitions that involve nearby goals rather than distant ones”.
>
> [4] Park, S., Ghosh, D., Eysenbach, B., & Levine, S. (2023). HIQL: Offline goal-conditioned RL with latent states as actions. Advances in Neural Information Processing Systems, 36, 34866-34891.
>
> In particular, we removed the notion of compounding error, see below.
>
> As for the sentence in Section 5.2, it was: “As Figure 2 shows, smaller values of $N$ tend to yield both higher sample efficiency and better performance in all environments tested. In particular, the best performance is observed for $N = 2$, corresponding to the setting in which the agent is conditioned on the next two upcoming goals, whereas the $N =$ final has the lowest performance and sample efficiency.”
>
> As already stated above, we have also modified it with:
>
> “As Figure 3 shows, learning a policy with smaller values of $N$ tends to perform equally or better in fewer interactions with the environment. In particular, the fastest improvement is observed for $N = 2$, corresponding to the setting in which the agent is conditioned on the next two upcoming goals, while it tends to be the slowest for $N =$ final.”
>
> **7) “only when using GCP-final the agent can prepare for the entire sequence of future goals.”**
>
> We hope the new Figure 1 now makes it clear that an agent which is not conditioned on the final goal cannot prepare for the sequence of future goals up to that final goal. In particular, Case 2 in Figure 1 accounts for the fact that a “myopic” agent ignores the sequence of goals to come. In Case 4, this is only when the two upcoming goals are the penultimate and the final goals that the agent becomes aware of this final goal.
>
> **8) “In such cases, the actor primarily learns to select actions that prepare the agent for reaching the first few intermediate goals, while the influence of more distant goals becomes negligible.“**
>
> Due to other comments from the reviewer about 5.2, the section has been widely reconsidered

---

> ### Author Response · Authors · 2026-01-01
>
> **Other claims:**
>
> **9) “However, the shorter horizon reduces sensitivity to compounding errors over time”**
>
> **Issue: What are the compounding errors? This is the only time this was discussed in the paper, and it is unclear what it is referencing or what evidence is used to make the claim**
>
> We thank the reviewer for helping us spot that our use of the notion of compounding error was inappropriate in this context. In model-based reinforcement learning, the errors made in predicting the next state from a model accumulate over the prediction horizon, which characterizes the compounding errors issue. Here we meant that a value function predicted with a goal-conditioned critic tends to be less accurate for states that are more distant to the goal. This phenomenon is clearly described in [4], although we are not 100% sure that their “signal-to-noise” ratio effect is the right explanation.
>
> So, as already mentioned above, we replaced the sentences as follows: “However, the shorter horizon improves the estimation of the value function [4], and the agent is more likely to have encountered training transitions that involve nearby goals rather than distant ones.”
>
> [4] Park, S., Ghosh, D., Eysenbach, B., & Levine, S. (2023). HIQL: Offline goal-conditioned RL with latent states as actions. Advances in Neural Information Processing Systems, 36, 34866-34891.

---

> ### Author Response · Authors · 2026-01-01
>
> **10) “This indicates that the critic trained with sac has only propagated the value from the first few goals of the sequence instead of all goals up to the final one.”**
> **Issue: This plot shows the difference in prediction error and does not directly show how value is propagated across subgoals. Thus, there are multiple ways this difference between the different values of N could have been observed. For example, the other methods (N=2,3) were able to reach the final goal faster, so they have had more time to learn the value function for a policy that goes through all the goals. This faster learning could just be due to beneficial hyperparameters or some factor other than value propagation, this claim is unsupported. Looking at value estimates and propagation during learning would be interesting and a possible route to explain the behavior. However, this experiment alone is insufficient to draw such a conclusion.**
>
> The reviewer did not mention which plot they were alluding to. We assume the reviewer meant Fig. 3(b). We agree with the reviewer that the differences observed in critic accuracy for different values of $N$ might in principle come from something else than value propagation. This point from the reviewer led us to widely reconsider Section 5.2. We made two main changes:
>
> - We realized that our main takeaway in Section 5.2 is that GCP-2 works more efficiently than variants where the policy is conditioned on more distant goals, no matter what the precise explanation is. Thus we rephrased our potential explanation in a more hypothetic tone and mentioned the explanations suggested by the reviewer as alternatives.
>
> - We further investigated our assumption in Appendix 7. In Figure 12, instead of plotting the critic value for each state of a trajectory as in Figure 3(b), we plot the evolution through training of the critic value at the start state and a coarse estimate of the Monte Carlo return from the same start state. Since only a single run is performed for each value of $N$, this study performed in Dubins Hallway is only qualitative but it shows that values converge faster to the unbiased Monte Carlo estimate of the return when $N$ is smaller. Interestingly, this plot also shows that expected values are larger for policies conditioned on more distant goals. This is likely because the critic considers more intermediate goals, hence more intermediate rewards. This might be another factor of slower convergence, as reaching a larger value from a random critic may take more time. But most importantly, the value from the critic reveals how many goals it takes into account, as a reward of +1 is associated with each intermediate goal. We believe the shown results corroborate our assumption that a value propagation issue is at play. Actually, in the xGC-MDP framework, reaching the goals in the prescribed order is mandatory. If a policy reaches the first and third subgoal while missing the second, it is rewarded only for the first one. Consequently, when the value estimate is around 3, the critic predicts that only the next 3 goals will be reached (ignoring discounting). For this reason, we believe that the value estimate provides an indication of how the reward of future goals has been propagated to the value of the current state.
>
> **There is another issue, where it is claimed that "In deterministic environments, the Monte Carlo return and Vπ coincide," but this is only true if the policy is deterministic. In this case, the policy is not deterministic, and the value function cannot be evaluated from a single episode. Furthermore, if a deterministic policy was created from a stochastic policy, then the value function would have been trained on the data from the stochastic policy, and there is no reason to assume it should match unless the policy is deterministic.**
>
> We agree with the reviewer. Our statement was wrong. We have revised the statement to: “In deterministic environments with deterministic policies a single episode return and $V^{\pi}$ coincide”. While we use SAC, which uses a stochastic policy, we empirically evaluated the learned policy over hundreds of trajectories. At the end of training, from the visualized start state to the final goal, all trajectories were nearly identical. In practice, the policy is deterministic in this setting. We also added a footnote to clarify this point.

---

> ### Author Response · Authors · 2026-01-01
>
> **11) “The agent does not differentiate between intermediate and final goals, leading it to decelerate before each goal to ensure it can stabilize there.”**
>
> **Issue: This is likely true, but there is no evidence for this claim. It would make sense that this is the desired behavior, but this behavior needs to be measured.**
>
> We agree with the reviewer. To solve the issue, we added velocity plots in former Figure 6 (now Figure 7), where SnakeMaze5 and GC-Cartpole are environments with terminal and non-terminal goals respectively. One can see in plots (e) in Figure 7 that SAC+HER+SEQ slows down before each intermediate goal in GC-Cartpole. We have updated the analysis of the results in Section 5.4 to account for this additional evidence.
>
> **12) “In SnakeMaze5, sac+her+seq instead reaches goals at high speed. However, this behavior makes it difficult to turn, resulting in bouncing off walls (Figure 6 (a)). As shown in Figure 6(b), the time the agent needs to reach the final goal is longer than with GCP-2.”**
>
> **Issue: Similar to the other statement, there is no direct evidence that it reaches the goals at higher speeds. The paths shown in Figure 6 (a) do not show velocity. The paths shown in Figure 6 (a) would be evidence that conditioning on the second goal helps the policy over just having the first goal, but not sufficient on its own. This is because, even without any goal information, a good policy should be able to navigate to the chosen goal as quickly as possible. This suggests (i.e., needs investigation) that two or more subgoals used to select actions may influence the policy (change the objective surface) to prefer a smooth path between goals, rather than simply seeking to reach a single goal as quickly as possible. This could be a route to understanding Figure 6 (a), but it is not the only one.**
>
> The additional plots in former Figure 6 (now Figure 7) should solve the reviewer’s point. It is now clear from the velocity plots in Figure 7(d) that the sac+her+seq agent reaches goals at high speed and then hits walls, resulting in abrupt speed changes, whereas the GCP-2 agent keeps its velocity more constant. We argue that this behavior arises from differences in the optimization objectives of the two methods. We hope that this new evidence, together with the clearer explanations provided in Figure 1 and Appendix A.7, will better convince the reviewer that our interpretation of these experimental results is correct.
>
> **Question: Were these paths smoothed in the plotting? They look unusually smooth for an RL agent.**
>
> Our agent is controlled with accelerations rather than positions, which tends to make trajectories smoother as the dynamics filters out the variations in the control signal. We suspect the reviewer expected more usual behavior corresponding to controlling the agent in position rather than in acceleration.

---

> ### Author Response · Authors · 2026-01-01
>
> **Requested Changes:**
>
> **All claims are either properly supported or scoped to the available evidence. For many of these claims, it would likely require new experiments that look beyond simple performance metrics. If ALL claims currently in the paper were only scoped to the available evidence, the paper would no longer have interesting results.**
>
> We have worked hard to address all the points where the reviewer pointed to us some lack of rigor in the expression of our claims, regarding metrics and statistical testing of our claims.  We have also provided more evidence (e.g. Velocities in Figure 7) when it was lacking. Again, we thank the reviewer for asking for more rigor, we believe our effort in this direction has helped us improve the paper a lot. We hope the reviewer will reward this effort with a better opinion of our work.
>
> **It is unclear why one would want to condition on a subgoal N>2 steps in the future and not include the subgoals between the current one and the Nth one. The value of N=2 is the only choice that actually chains together subgoals. It should be made clear why N>2 is even considered or could be beneficial. The stated reasons (as discussed above) do not follow.**
>
> As discussed in more detail above, conditioning the policy on the full list of goals raises additional issues when the length of this list varies from one trajectory to another. Conditioning a policy on a variable length object can be addressed in different ways: building a fixed length embedding out of a dedicated architecture such as a Transformer or a RNN, or using a maximal length conditioning vector, then masking unused bits for inputs shorter than the vector and ignoring extra bits for inputs longer than the vector. Some preliminary experiments convinced us that these approaches required a paper on its own. We thus believe studying these approaches is beyond the scope of the paper.
>
> Besides, we believe our approach provides an elegant and frugal solution to these concerns, we hope that the clarification brought by Figure 1 and some rephrasing will have convinced the reviewer that this approach has its own merits.
>
> **Important but not critical: It should be made clear why we are looking at planners that provide bad subgoals. For example, in the walker problem, the sequence of subgoals is just a series of vectors to move straight ahead. Why not have just a single goal for this problem to reach the final position? There are many examples (and I have personal experience) that make the humanoid move to different positions and require the agent to change directions. What does providing a single**
>
> Could the reviewer kindly provide the end of the last sentence which was missing in the review? About the specific case of the GC-Humanoid-Walk experiments, given our limited computational budget, we simplified the standard goal-reaching setup. Specifically, the humanoid always starts oriented towards the goal, so it does not need to handle changes in orientation. So the straight line of subgoals we provide in these experiments is actually a good enough plan, though the distance between goals was not optimized. For the more general question of using a fixed plan without any guarantees on the quality of this plan, we refer the reviewer to our discussion with Reviewer hR8T who raised the point.
>
> **Additional Comments: It should be stated how $k_1$  and $k_2 are chosen in the method.**
>
> $k_1 = \min(t_1, t_2)$ and $k_2 = \max(t_1, t_2)$. $t_1, t_2$ are taken randomly between the current time step and the time at the end of the trajectory. We rephrased a sentence in the Hindsight Relabeling paragraph of Section 4.2, pp. 8-9, to clarify this point.

---

> ### Comment · Reviewer_teVv · 2026-02-02
>
> >From Figure 3: "Each evaluation is performed 10 times, with results reported as the mean and 95% confidence interval (computed with bootstrapping) over 30 seeds. Stars denote statistically significant differences (Welch T-test, p-value <0.05) between GCP-2 and the algorithm of the corresponding color."
>
> Bootstrap intervals are reliable when the sample data distribution looks similar to the actual data distribution. Per each run of the algorithm, there is a 5% change you will sample a performance that lies in the (0,2.5) or (97.5,100) percentile. With 30 seeds (30 runs), that is approximately a 21.5% chance that none of the runs come from this range. RL algorithms are known to randomly fail, which have a large impact on the average learning curve. The percentile bootstrap cannot correct asses the uncertainty when samples that are far away from the mean are missed. How likely is the algorithm to fail (1/10, 1/30, 1/100, 1/1000)? Without knowing this, it is hard to determine how much to trust the percentile bootstrap intervals. Generally, one does not trust bootstrap techniques with fewer than 50 samples, unless you somehow already know you can trust them.
>
> >As we are only performing one-to-one algorithm comparisons, we do not incorporate Bonferroni correction, but we are ready to change this if the reviewer feels it is necessary.
>
> This is not a correct comparison. Each comparison is made independently, but there are 10 comparisons being made, and one algorithm is being compared against 3 others, which makes for a total of 30 comparisons. This means the upper bound on the probability of failure for any one ofthese statistical tests is 30*5% > 100%. You can say that these are all done pointwise, that is fair (if it is made clear that some of them are likely to be wrong), but is a very weak comparison and it is not clear what should be drawn from these tests.
>
> Personally, I think you could make an easier statistical test if you just provide the learning curve to show what performance looks like, and instead look comparing E[L] for each algorithm, where L is a random variable representing the time it takes for the algorithm to find a policy that scores above some threshold of performance. The threshold choice is up to you, but probably reasonable to justify with something like a success rate $\ge 99$%. You would need to change the experiment to test the policy every X episodes to check for the threshold passing, but you do not have to compare at 10 points on the learning curve. This metric also directly captures a measure of sample efficiency. This is just a suggestion; it is your paper, and I only care about the claims being well supported.
>
> >About hyper-parameters, our approach is the following. The hyper-parameters used for the SAC backbone in all methods were mainly taken from the original SAC paper [2]. We only increased the neural network size for more complex tasks, as in [3]. We added a sentence in Appendix A.3 to account for this fact.
>
> So this means we have no information about whether the hyperparameter choice happens to favor one method over another. As such, all conclusions should be scoped to say that they only pertain to these specific hyperparameter choices and it is unclear if these results would carry over to other hyperparameters.

---

> ### Comment · Reviewer_teVv · 2026-02-02
>
> >We agree that we did not define a stability metric and that, although sample efficiency is a well-understood notion in the RL literature that does not need to be defined any more (see e.g. [4] and many others), our use of it was quite vague in the above statement. In the revised version, we replaced the above sentence with: ... “2) By evaluating policies trained with an off-policy actor-critic algorithm on both the standard goal-conditioned MDP framework and ours in these tasks, we show that learning policies conditioned on the next two goals generally require less interaction data than all the other types to reach the same level of performance or better.”
>
> I agree that the general notion of sample efficiency (higher performance with less data) is well understood; the problem is that it is unclear which metric is used to make the determination. Is is acheiving X level of performance in the fewest steps? Is it getting higher performance in L steps? These two are specific and measurable (thus testable) definitions of sample efficiency. Without being specific, it is up to interpretation whether the method is more sample efficient. Not to mention that something measurable is required to use statistical tests to determine whether there is enough evidence for it.
>
> >“Our method, GCP-2, consistently achieves the highest or comparable success rates in all environments. It is the only method whose success rate starts to climb over 1% after 2M steps in the GC-Humanoid-Walk task (getting above 40% after 10M steps), and gets close to 100% in Dubins Hallway in less than 150K steps.”
>
> These are getting closer to being specific and measurable. These are fine observations to make and comment on, but to be used as direct evidence for a claim, there should be a statistical test used to verify them. Side note: if you are choosing values now to compare against, you need new data, or the result will be biased because YOU have already looked at the data.

---

> ### Comment · Reviewer_teVv · 2026-02-02
>
> >We have worked hard to address all the points where the reviewer pointed to us some lack of rigor in the expression of our claims, regarding metrics and statistical testing of our claims. We have also provided more evidence (e.g. Velocities in Figure 7) when it was lacking. Again, we thank the reviewer for asking for more rigor, we believe our effort in this direction has helped us improve the paper a lot. We hope the reviewer will reward this effort with a better opinion of our work.
>
> I do appreciate the work; both my understanding and the paper's quality have improved. I believe there is still an issue with the claims (see other comments). The claims are probably true, as they follow what is intuitive and established trends, e.g., optimizing shorter-horizon problems is faster than longer-horizon problems, which is similar to using a smaller N. However, the experiments used contain methodological errors that cannot justify the conclusions. In addition to fixing these errors, I think improving the writing would be the best way to improve the paper.
>
> >Could the reviewer kindly provide the end of the last sentence, which was missing in the review?
>
> Oooops, sorry! It ends as follows. You can ignore the incomplete sentence.
>
> *continuing humanoid discussion*: My point with the humanoid experiment is that I'm not sure what it teaches us about the GCP-N method. The subgoals do not require smoothness between them like SnakeMaze5, and it certainly does not need the final goal to achieve good behavior; it only needs to know how to run forward, e.g., specify the goal in velocity space, not position space. For SAC+HER and SAC+HER+SEQ the goal specifications are very poor choices. It should be made clear what is hoped to be learned from each environment/experiment.

---

> ### Comment · Reviewer_teVv · 2026-02-02
> **Question about necessity of GCP-final**
>
> I understand that, in principle, the GCP-final is the only one with an objective that optimizes for the full sequence of behavior goals, and using N<final could lead to a bad policy. However, I do not see an experiment that demonstrates that it is ever necessary. Is there an example that can be provided to show that GCP-N, N<final, leads to a bad policy, i.e., conditioning on the full sequence is necessary? Or even one where N>2 is required? Right now, it looks like there is a hidden assumption that the behavior goal planner provides a sequence of goals where N=2 is sufficient. What would it look like where N=2 is not sufficient? I think this is actually a critical piece missing in understanding the limitations of the approach. This could be achieved by creating a small test environment to demonstrate the necessity of N>2.

---

> ### Author Response · Authors · 2026-02-12
>
> Again, we thank the reviewer for keeping pushing us towards more accurate statements in our paper. We respond to all the Reviewer’s points below:
>
> >Bootstrap intervals [...]. Generally, one does not trust bootstrap techniques with fewer than 50 samples, unless you somehow already know you can trust them.
>
> We agree with the reviewer that statistical tests based on bootstrap confidence intervals generally require a large number of samples. To solve the Reviewer’s concern, we have updated all our results and tests using 50 samples.
>
>
> >> *“[...] we do not incorporate Bonferroni correction [...] ”
>
> >This is not a correct comparison. Each comparison is made independently, but there are 10 comparisons being made, and one algorithm is being compared against 3 others, which makes for a total of 30 comparisons. This means the upper bound on the probability of failure for any one of these statistical tests is 30*5% > 100%. You can say that these are all done pointwise, that is fair (if it is made clear that some of them are likely to be wrong), but is a very weak comparison and it is not clear what should be drawn from these tests.
>
> We agree with the reviewer that with many independent pointwise comparisons, the probability of at least one false positive is high. However, the probability that all tests are simultaneously wrong is orders of magnitude lower. We have kept these results as we believe they offer useful insight when comparing the learning curves, but we do not base any claim directly on them, as detailed in the following responses.
>
>
> >Personally, I think you could make an easier statistical test if you just provide the learning curve to show what performance looks like, and instead look at comparing E[L] for each algorithm, where L is a random variable representing the time it takes for the algorithm to find a policy that scores above some threshold of performance. The threshold choice is up to you, but probably reasonable to justify with something like a success rate %. You would need to change the experiment to test the policy every X episodes to check for the threshold passing, but you do not have to compare at 10 points on the learning curve. This metric also directly captures a measure of sample efficiency. This is just a suggestion; it is your paper, and I only care about the claims being well supported.
>
>
> We thank the reviewer for suggesting a clear sample efficiency metric. We have now included it: the average number of steps needed to reach 99% success rate (SR), with statistical tests between methods. A note is that this can only be calculated when all runs eventually reach 99% SR; otherwise, instead of sample efficiency, the statistical test evaluates the final SR. We have updated the paper with new tables for each figure with learning curves. This table shows both final performance and the time to reach 99% SR, along with statistical tests to support our claims.
>
> >> About hyper-parameters, our approach is the following. The hyper-parameters used for the SAC backbone in all methods were mainly taken from the original SAC paper [2]. We only increased the neural network size for more complex tasks, as in [3]. We added a sentence in Appendix A.3 to account for this fact.
>
> >So this means we have no information about whether the hyperparameter choice happens to favor one method over another. As such, all conclusions should be scoped to say that they only pertain to these specific hyperparameter choices and it is unclear if these results would carry over to other hyperparameters.
>
> We agree that performance of each agent depends on the choice of hyper-parameters and that each of them could be affected differently. To clarify the scope of our claims, we have revised the manuscript:
>
> "All these agents rely on SAC+HER with the same hyper-parameters (see Appendix A.3), conclusions are therefore only drawn under these configurations, without specific fine-tuning for each agent.."

---

> ### Author Response · Authors · 2026-02-12
>
> >> We agree that we did not define a stability metric and that, although sample efficiency is a well-understood notion in the RL literature that does not need to be defined any more (see e.g. [4] and many others), our use of it was quite vague in the above statement. In the revised version, we replaced the above sentence with: ... “2) By evaluating policies trained with an off-policy actor-critic algorithm on both the standard goal-conditioned MDP framework and ours in these tasks, we show that learning policies conditioned on the next two goals generally require less interaction data than all the other types to reach the same level of performance or better.”
>
> >I agree that the general notion of sample efficiency (higher performance with less data) is well understood; the problem is that it is unclear which metric is used to make the determination. Is it achieving X level of performance in the fewest steps? Is it getting higher performance in L steps? These two are specific and measurable (thus testable) definitions of sample efficiency. Without being specific, it is up to interpretation whether the method is more sample efficient. Not to mention that something measurable is required to use statistical tests to determine whether there is enough evidence for it.
>
> We agree with the reviewer on this point. As previously explained, we have now incorporated the metric they proposed, which assesses which method achieves "a 99% success rate in the fewest steps." As a result, every claim in our article is now statistically supported using either the final performance or this new sample efficiency metric.
>
> >> “Our method, GCP-2, consistently achieves the highest or comparable success rates in all environments. It is the only method whose success rate starts to climb over 1% after 2M steps in the GC-Humanoid-Walk task (getting above 40% after 10M steps), and gets close to 100% in Dubins Hallway in less than 150K steps.”
>
> >These are getting closer to being specific and measurable. These are fine observations to make and comment on, but to be used as direct evidence for a claim, there should be a statistical test used to verify them. Side note: if you are choosing values now to compare against, you need new data, or the result will be biased because YOU have already looked at the data.
>
> The descriptions regarding the timing of performance improvements (e.g., "starts to climb after 2M steps") were included as qualitative observations to illustrate trends. Our core claim is now focused solely on superior final success rate, which is validated by the statistical tests that we have reported.

---

> ### Author Response · Authors · 2026-02-12
>
> >Ah ok, I believe I better understand the argument now (added text in Section 4.2 helped). However, I think the text is unclear. My understanding is that GCP-N optimizes for following a sequence of N behavior goals and uses a policy conditioned on the next goal and the N-th goal. The policy is then used in an environment with K>N behavior goals, and once a behavior goal is achieved, the problem becomes a K-1 behavior goal problem. So the GCP-N method should work well when 1) the agent can follow a sequence of N behavior goals, 2) the agent only needs to be myopic with respect to N behavior goals, and 3) the policy does not cause the agent to be in an unfavorable state when achieving a behavior goal and the next fg switches to the next behavior goal. With N=2 the policy is only optimized for following a sequence of two behavior goals and thus should provide a smooth transition between behavior goals since actions are always selected to enable the agent to get to the next behavior goal. However, when the last goal is more than N behavior goals away, the policy may take actions that prevent the agent from reaching it (violating assumption 2). Assuming my understanding is correct, I think Sections 2 and 4 should be rewritten to make it clearer and more concise.
>
> Let us decompose the points from the reviewer:
>
> **My understanding is that GCP-N optimizes for following a sequence of N behavior goals and uses a policy conditioned on the next goal and the N-th goal. The policy is then used in an environment with K>N behavior goals, and once a behavior goal is achieved, the problem becomes a K-1 behavior goal problem.**
>
> Yes.
>
> **So the GCP-N method should work well when 1) the agent can follow a sequence of N behavior goals, 2) the agent only needs to be myopic with respect to N behavior goals, and 3) the policy does not cause the agent to be in an unfavorable state when achieving a behavior goal and the next fg switches to the next behavior goal. With N=2 the policy is only optimized for following a sequence of two behavior goals and thus should provide a smooth transition between behavior goals since actions are always selected to enable the agent to get to the next behavior goal. However, when the last goal is more than N behavior goals away, the policy may take actions that prevent the agent from reaching it (violating assumption 2).**
>
> For (3), we would suggest a small clarification: it should say "and the next fg switches to the next N^th goal"  unless we are in the specific case of GCP-2. Also, we believe property (3) actually directly depends on (1) and (2).
>
> **“Assuming my understanding is correct, I think Sections 2 and 4 should be rewritten to make it clearer and more concise.”**
>
> We agree with the reviewer that the properties they pointed out are an interesting way of describing GCP-N agents. We have revised the concluding text of Section 4.3 to include these descriptions. However, the main text in Sections 4.2 and 4.3 aims to formalize how GCP-final and GCP-N are learned, linking the idea "the method optimizes for the next N goals" to how this is implemented and formalized with an MDP. We believe these points are fundamental before further analysis. Here is the concluding text we have added:
>
> […] For a GCP-N agent to achieve good performance, it must (1) learn a policy that can always reach the next $N$ goals, and (2) operate in a problem setting where optimizing only the next $N$ goals is sufficient. Failure to satisfy (2) can occur in two ways: the agent may reach an intermediate goal in a configuration that is not compatible with reaching the final goal leading to failure, or it may complete the full sequence but with suboptimal behaviors. In contrast, GCP-final does not require property (2), since it directly optimizes the full sequence. However, for GCP-final, property (1) may be harder to achieve. Shorter horizons improve value function estimation [1], and during training the agent is more likely to observe transitions involving nearby goals than distant ones.
>
> [1] Park, S., Ghosh, D., Eysenbach, B., & Levine, S. (2023). HIQL: Offline goal-conditioned RL with latent states as actions. Advances in Neural Information Processing Systems, 36, 34866-34891.

---

> ### Author Response · Authors · 2026-02-12
>
> >In addition to fixing [claim-related] errors, I think improving the writing would be the best way to improve the paper. My point with the humanoid experiment is that I'm not sure what it teaches us about the GCP-N method. The subgoals do not require smoothness between them like SnakeMaze5, and it certainly does not need the final goal to achieve good behavior; it only needs to know how to run forward, e.g., specify the goal in velocity space, not position space. For SAC+HER and SAC+HER+SEQ the goal specifications are very poor choices. It should be made clear what is hoped to be learned from each environment/experiment.
>
> We agree with the reviewer that a velocity-space goal specification would be an efficient way to solve the humanoid task. However, our focus is on settings where goal chaining is essential. We intentionally use a position-based goal formulation in the humanoid environment to study high-dimensional state and action spaces, together with the additional challenge that the agent must reach each intermediate position in a configuration from which it can successfully proceed to the next goal. As shown in our results, the SAC+HER+SEQ baseline, which treats each goal independently, achieves a very low success rate. We also include a qualitative failure example in which this baseline reaches the first goal with an unstable posture (e.g., legs too far apart), leading to a fall.
> The reviewer’s comment highlighted that our motivation for the choice of evaluation environments was not clearly stated. To address this, we added a paragraph at the beginning of Section 5.1 that explicitly explains the purpose of each environment and what we aim to learn from these experiments. The paragraph is as follows:
> “We evaluate the methods in four environments, each highlighting different goal-chaining challenges. In Dubins Hallway, ignoring future goals can lead to collisions with walls, resulting in failure. PointMaze and Cartpole illustrate how terminal and non-terminal conditions, respectively, can affect performance in sequences of goal following. Finally, GC-Humanoid-Walk tests scalability to high-dimensional action and state spaces, where each intermediate goal must be reached in a sufficiently balanced configuration to allow continued locomotion toward the final goal.”
>
>
> >I understand that, in principle, the GCP-final is the only one with an objective that optimizes for the full sequence of behavior goals, and using N<final could lead to a bad policy. However, I do not see an experiment that demonstrates that it is ever necessary. Is there an example that can be provided to show that GCP-N, N<final, leads to a bad policy, i.e., conditioning on the full sequence is necessary? Or even one where N>2 is required? Right now, it looks like there is a hidden assumption that the behavior goal planner provides a sequence of goals where N=2 is sufficient. What would it look like where N=2 is not sufficient? I think this is actually a critical piece missing in understanding the limitations of the approach. This could be achieved by creating a small test environment to demonstrate the necessity of N>2.
>
> Let us decompose the points from the reviewer:
>
> **Environments where only optimizing the full sequence is necessary.**
>
> Reviewer hR8T raised a similar point. In Appendix A.4, we provide a tabular example illustrating that for specific combinations of planner, environment dynamics, and goal-space specifications, successful task completion requires an agent to plan over the entire goal sequence.
>
> **Is there a hidden assumption that the planner always provides a sequence of goals where N=2 is sufficient ?**
>
> We agree that in our current benchmarks, planning for only the next two goals is often sufficient to achieve high success rates, as shown by the performance of GCP-2. This observation, however, does not imply that such a horizon is sufficient to reach optimal performance in any of them. For instance, the oscillations in the speed profile of Cartpole suggest that optimization over a longer horizon could yield better behavior. Consequently, more robust optimization of these longer-horizon objectives might eventually outperform GCP-2 in most tested environments.

---

### Decision · Action_Editor_eYL5 · 2026-02-25

**Recommendation:** Accept as is

**Audience:**

Yes

**Audience Explanation:**

Goal-conditioned RL is of interest to many RL researchers.

**Claims And Evidence:**

Yes

**Claims Explanation:**

The authors and reviewers have extensively discussed the claims in the work, authors have incorporated improvements and the claims are appropriately scoped and supported.